# Crystal structure of lipid A disaccharide synthase LpxB from *Escherichia coli*

Heather O. Bohl[1], Ke Shi[1], John K. Lee[1,2] & Hideki Aihara[1]

Most Gram-negative bacteria are surrounded by a glycolipid called lipopolysaccharide (LPS), which forms a barrier to hydrophobic toxins and, in pathogenic bacteria, is a virulence factor. During LPS biosynthesis, a membrane-associated glycosyltransferase (LpxB) forms a tetra-acylated disaccharide that is further acylated to form the membrane anchor moiety of LPS. Here we solve the structure of a soluble and catalytically competent LpxB by X-ray crystallography. The structure reveals that LpxB has a glycosyltransferase-B family fold but with a highly intertwined, C-terminally swapped dimer comprising four domains. We identify key catalytic residues with a product, UDP, bound in the active site, as well as clusters of hydrophobic residues that likely mediate productive membrane association or capture of lipidic substrates. These studies provide the basis for rational design of antibiotics targeting a crucial step in LPS biosynthesis.

[1] Department of Biochemistry, Molecular Biology, and Biophysics, University of Minnesota Twin Cities, Minneapolis, MN 55455, USA. [2] Present address: Bristol-Myers Squibb, Redwood City, CA 94063, USA. Correspondence and requests for materials should be addressed to H.A. (email: aihar001@umn.edu)

Antibiotic-resistant bacteria are an increasing threat to human health[1]. The development of new antibiotics to fight resistant strains can be aided by the characterization of new targets[2]. Most Gram-negative bacteria have an outer membrane with an outer leaflet composed of lipopolysaccharide (LPS), which contains a lipid A membrane anchor, core oligo-saccharide, and O-antigen polysaccharide[3]. LPS creates a barrier to hydrophobic toxins, increasing Gram-negative resistance to many antibiotics[3,4]. Resistant Gram-negative bacteria, such as *Pseudomonas aeruginosa*[5], *Acinetobacter baumannii*[6], and *Klebsiella pneumonia*[7], are an increasing medical problem[1]. Moreover, LPS and lipid A contribute to the mortality and morbidity of Gram-negative infections, because they can over-stimulate the host immune system and cause septic shock[3]. As LPS is critical for the viability of most Gram-negative bacteria[3], enzymes that are essential for the synthesis of a functional LPS molecule are attractive targets for antibiotics.

LpxB is a glycosyltransferase in the Raetz (lipid A synthesis) pathway that catalyzes nucleophilic attack of the 6′-hydroxyl of lipid X (**1**) on the anomeric carbon of UDP-diacyl-glucosamine (UDP-DAG) (**2**) to form β(1–6)-tetraacyl-disaccharide 1-phosphate (lipid A disaccharide) (**3**) (Fig. 1)[8–12]. LpxB is essential for growth of *Escherichia coli* and is among the most highly conserved enzymes in the Raetz pathway[13,14]. LpxB was first identified from a temperature-sensitive mutant of *E. coli* (*pgsB1*) that was shown to accumulate lipid X and lipid Y (a palmitoylated form of lipid X) at 42 °C[15,16]. LpxB was fully purified and enzymatically characterized by Metzger and Raetz[10] who showed that LpxB is a membrane surface active enzyme.

Of the enzymes present in the canonical Raetz pathway, seven have been structurally characterized (LpxA[17], LpxC[18], LpxD[19], LpxH[20,21], LpxK[22], WaaA[23], and LpxM[24]) and two of them have not (LpxB and LpxL). In addition, LpxI (a non-homologous alternative to LpxH) has been structurally characterized by X-ray crystallography[25,26]. Antibiotic lead compounds that target LpxC[27–33], LpxH[34], LpxD[35], and LpxA[35,36] have been developed, but none have been reported for LpxB.

Here we present the first crystal structures of a soluble variant of *E. coli* LpxB in the apo form and bound to UDP, which reveals a novel glycosyltransferase-B (GT-B) dimer wherein the C-terminal tail of one subunit completes the fold of the other subunit. We further show importance of the LpxB dimerization via C-terminal swapping through functional analyses. This structure provides a basis for rational and computational design of antibiotic lead compounds targeting this essential step in LPS synthesis.

## Results

**Overall structure of LpxB.** In order to crystallize LpxB, a soluble form of the enzyme was generated. We utilized the Phyre2 server[37] to create a homology model of LpxB based on UDP-N-acetylglucosamine 2-epimerase from *Thermus thermophilus* HB8 (PDB: 1V4V). We compared the homology model with the structure of MurG (PDB: 1F0K and 1NLM)[38,39], which was found to have a hydrophobic patch likely involved in membrane association[39]. We identified, and mutated to Ser, six Val and Leu residues (V66, V68, L69, L72, L75, and L76) in a nearby region of the LpxB homology model. These mutations improved solubility and yield of LpxB, and the LpxB with all six mutations (LpxB6S) showed the least aggregation on a size exclusion column (Supplementary Fig. 1). LpxB6S was used for initial crystallization screening and optimization. An additional mutation (M207S) was added for selenomethionine derivative crystallization giving LpxB7S (Table 1).

LpxB6S was crystallized with one molecule in the asymmetric unit. However, the biological assembly is a highly intertwined, C-terminally swapped homodimer. Each half of the LpxB dimer consists of two Rossmann-like domains with a parallel β-sheet core sandwiched between α-helices (Fig. 2 and Supplementary

**Fig. 1** LpxB reaction. LpxB catalyzes the nucleophilic attack of the 6′-hydroxyl of lipid X (**1**) on the anomeric carbon of UDP-DAG (**2**) with UDP as the leaving group[9]. As for other inverting GT-B enzymes, this reaction is thought to proceed by an SN2 mechanism[40,41]

**Table 1 Diffraction and refinement statistics**

| | LpxB7S (PDB: 5W8S) | LpxB7S SeMet (SAD data) | LpxB7S+UDP (PDB: 5W8X) | LpxB6S (PDB: 5W8N) |
|---|---|---|---|---|
| *Data collection* | | | | |
| Space group | P 3₂ 2 1 | P 6₄ | P 3₂ 2 1 | P 3₂ 2 1 |
| Cell dimensions | | | | |
| $a, b, c$ (Å) | 68.02, 68.02, 155.1 | 111.2, 111.2, 74.19 | 67.81, 67.81, 153.9 | 67.78, 67.78, 154.50 |
| $\alpha, \beta, \gamma$ (°) | 90, 90, 120 | 90, 90, 120 | 90, 90, 120 | 90, 90, 120 |
| Resolution (Å) | 155.14–2.10 (2.16–2.10) | 96.29–3.43 (3.71–3.43) | 153.88–1.98 (2.03–1.98) | 19.57–2.02 (2.07–2.02) |
| $R_{merge}$ | 0.066 (0.997) | 0.097 (1.926) | 0.063 (1.272) | 0.112 (1.686) |
| $I / \sigma I$ | 13.5 (1.6) | 21.6 (1.9) | 13.2 (1) | 14.2 (1.2) |
| Completeness (%) | 99.4 (99.9) | 100 (100) | 99.1 (97.8) | 99.6 (97.0) |
| Redundancy | 3.5 (3.6) | 10.4 (10.4) | 4.1 (3.7) | 6.5 (6.1) |
| *Refinement* | | | | |
| Resolution (Å) | 55.07–2.10 (2.15–2.10) | Not fully refined | 54.86–1.98 (2.05–1.98) | 19.57–2.02 (2.09–2.02) |
| No. reflections | 24788 (1737) | NA | 26651 (2153) | 27680 (2650) |
| $R_{work} / R_{free}$ | 0.1951/0.2226 (0.3344/0.3789) | NA | 0.1990/0.2309 (0.3512/0.3278) | 0.1872/0.2129 (0.3043/0.3439) |
| No. non-hydrogen atoms | 2953 | NA | 2964 | 3043 |
| Protein | 2,800 | NA | 2,817 | 2,859 |
| Ligand/ion | 3 | NA | 25 | 0 |
| Water | 150 | NA | 122 | 184 |
| *B*-factors | 45.6 | NA | 46.12 | 46.06 |
| Protein | 45.39 | NA | 46.06 | 45.80 |
| Ligand/ion | 51.25 | NA | 52.67 | |
| Water | 49.44 | NA | 46.15 | 50.20 |
| R.m.s. deviations | | | | |
| Bond lengths (Å) | 0.003 | NA | 0.003 | 0.005 |
| Bond angles (°) | 0.507 | NA | 0.572 | 0.623 |

Statistics for highest resolution shell are shown in parentheses. Each data set corresponds to one crystal
NA, not applicable

Fig. 2). Both polypeptides within the LpxB dimer contribute to the fold of each domain. The first Rossmann-like domain is composed of the N-terminal half of one subunit (residues 1–166) and the last helix (369–382) from the C-terminus of the other subunit. This domain is connected via short antiparallel helical linkers to the second domain, which is composed of residues 177–295 of the first subunit and 296–345 of the second subunit. Although the tertiary structure of LpxB is typical for the GT-B family, the quaternary structure is unique[40,41]. LpxB forms a two-fold symmetric dimer in which the polypeptide chains crossover at helix 14 around the dyad axis to swap the remainder of the polypeptide chain. The reciprocally swapped C-terminal segments wrap around the opposing subunit in an interlocked arrangement. These extensive interactions bury a total surface area of 12,650 Å², forming an LpxB dimer with an overall 'bowtie'-shape. Supplementary Figure 2 presents the isolated monomer and a 2D depiction[42] of the dimer.

**UDP binding site**. The ligand-bound structure of LpxB was obtained by soaking crystals with 10 mM UDP-*N*-acytlglucosamine (UDP-GlcNAc), a mimic of one of the reactants, UDP-DAG (Fig. 3). Only the UDP portion of the molecule is resolved in the electron density map (Fig. 3b), suggesting that the GlcNAc moiety is highly flexible or that UDP-GlcNAc was hydrolysed during soaking as has been observed before for nucleotide-charged sugars soaked or co-crystallized with GT-B enzymes[43–45]. The uracil base binds in a hydrophobic pocket formed by L197, P198, P231, and V233, which is on the second Rossmann-fold domain and facing the deep inter-domain cleft. The P231 carbonyl oxygen also hydrogen-bonds with N3 of uracil and the G199 amide nitrogen hydrogen-bonds with the O4 carbonyl of uracil. In addition, two water molecules connect active site residues to uracil: the first water connects the G199 carbonyl oxygen and the

V233 amide nitrogen to O4 of uracil, and the second water connects the G261 amide nitrogen to the O2 carbonyl of uracil. The ribose 2′ and 3′ hydroxyls both hydrogen-bond with the E281 side chain. The α- and β-phosphates hydrogen-bond with the S200 and T277 side chains, respectively. R201 also contacts both the α- and β-phosphates of UDP, albeit at a slightly longer distance (3.8 Å). R201 was shown to be critical for LpxB enzymatic activity, and this Arg side chain likely has a catalytic role in stabilizing the negatively charged transition state of the UDP leaving group[10].

**Membrane association of LpxB**. We assessed the activity of solubilized LpxB (LpxB6S) by in vitro enzymatic assays as well as by genetic knockout and complementation. In vitro activity assays suggested that the ability of LpxB6S to catalyze the reaction of Triton X-100-solubilized substrates was completely abolished. Even after the reactions proceeded for over 17 h, the amount of UDP released was not above that of a zero enzyme control. Thus, the activity of LpxB6S was below the detection limit under these conditions (Table 2). In contrast, the specific activity of wild-type LpxB was measured at 6.17 ± 0.53 μmol min⁻¹ μmol⁻¹ (Table 2). However, when the substrates were solubilized with 0.9 M 3-(1-Pyridinio)-1-propanesulfonate (NDSB 201), residual lipid A disaccharide synthesis was observable for LpxB6S by thin layer chromatography (TLC) (Supplementary Fig. 3), but LpxB6S activity remained below the detection limit for UDP-release assays even with the addition of NDSB 201 (Table 2). These results demonstrate that LpxB6S remains catalytically competent and suggest that this solubilized form of the enzyme fails to extract lipid substrates from detergent micelles. To probe which of the residues mutated in LpxB6S are more important, we tested the in vitro activity of less mutated versions of LpxB. We tested two sets of triple mutants: V66SV68SL69S (VVL), which are

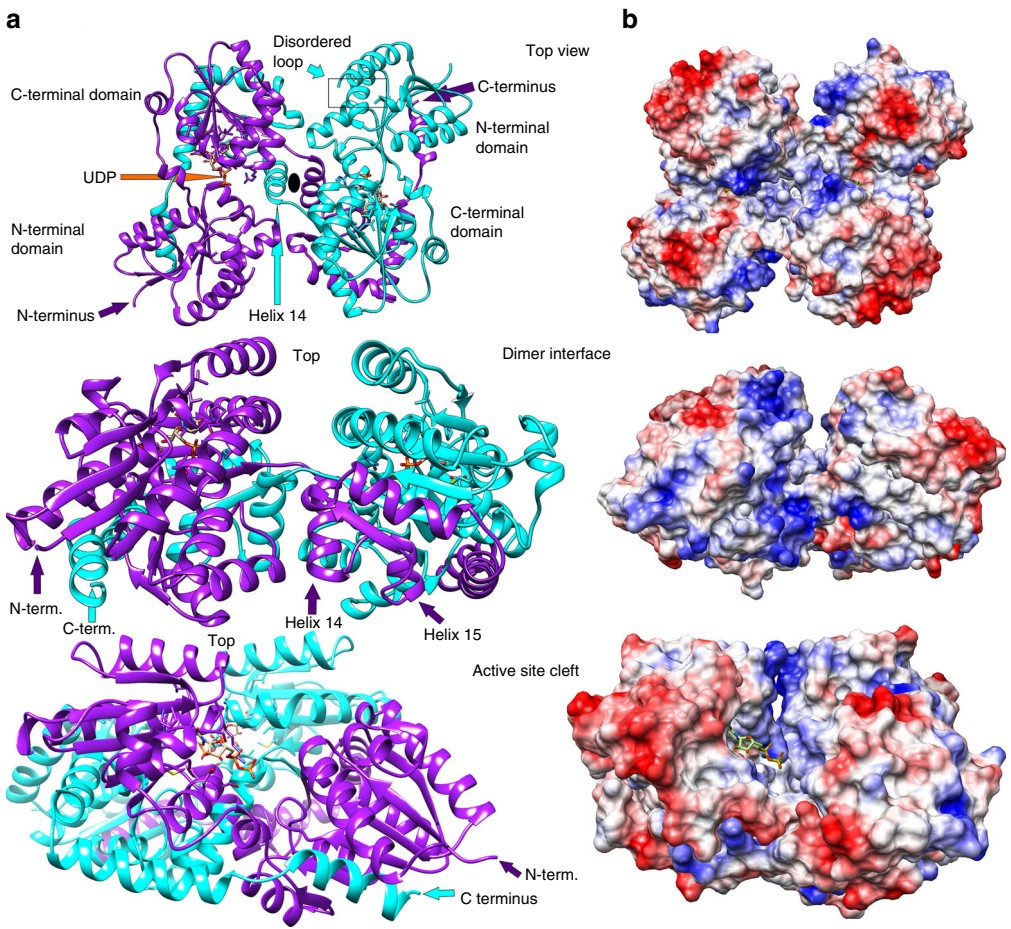

**Fig. 2** Overall structure of LpxB. **a** The overall structure of LpxB7S shows that LpxB forms a dimer in which the C-terminus of one subunit completes the fold of the opposite subunit. The polypeptides are exchanged at helix 14. Each domain forms a Rossmann-like fold with a parallel β-sheet surrounded by α-helices. UDP is bound to the C-terminal domain showing the position of the UDP-binding pocket where the sugar donor substrate, UDP-DAG, binds. Lipid X likely binds to the N-terminal domain which includes the predicted catalytic base D98[10]. LpxB appears to be in an open conformation. Catalysis may involve partial closing of the cleft between the domains by a hinge-like movement of the N-terminal domain similar to conformational changes observed for MurG[38], PimA[50], and other GT-B enzymes[40]. **b** Columbic surface rendering (blue for positive and red for negative) corresponding to the adjacent ribbons in **a**. The surface near the dimer interface is highly basic and is likely involved in membrane association

located in the disordered loop (residues 62–71), and L72SL75SL76S (LLL), which is located in the following helix (Fig. 2). Both triple mutants were active with detergent solubilized substrates, but they showed significantly different activities (Table 2): the specific activity of LpxBVVL was measured at 0.053% of wild-type while that of LpxBLLL was 0.0010% of wild type (Table 2). The lower activity of the LLL mutant suggested that this later group of residues is more important for substrate binding. However, double mutants carrying any pair of L72S, L75S, and L76S retained more activity than LpxBVVL with LpxBL72SL75S showing the greatest decrease in activity (1.74% of wild type versus ~ 1/3 of wild type for the other two) (Table 2). These data support the hypothesis that the hydrophobicity of this surface patch is required for extraction of lipid substrates from detergent micelles as greater decrease in overall hydrophobicity, rather than mutation of a specific residue, correlated with decrease of activity (Table 2).

**Activities of previously characterized mutants.** Wild-type LpxB and two mutants (N316A and R201A) made for this study have been previously characterized by Metzger and Raetz[10]. However, the reaction conditions utilized for comparison of specific activities in this study were quite different from those utilized

previously[10]. Most importantly, the substrate concentrations were 31 μM UDP-DAG and ~ 130 μM lipid X in this study (versus 600 and 400 μM in ref. [10]), and reactions were run at ambient temperature (~ 21 °C) in this study (versus 30 °C in ref. [10]). Thus, the specific activity of the wild type was much lower (~ 1,200-fold) under the conditions of this study. However, the relative activities were consistent: the specific activity of LpxBN316A was measured at 0.34% of wild type here and at 0.1% previously, and the specific activity of R201A was measured at 0.0051%, consistent with the < 0.01% reported previously[10] (Table 2). The comparatively high activity of LpxBN316A is consistent with the observed environment of N316 in the crystal structure. Rather than being positioned to interact with the substrates, the N316 side chain hydrogen-bonds with the backbone of residues 323 and 324, and appears to stabilize the positioning of helix 15 (Fig. 4b). LpxBF298EN316A (LpxBFN) showed further reduced activity (0.0078% of wild-type) (Table 2). Similar to N316, F298 is from the swapped C-terminal segment, and it makes cation-π stacking in *trans* with the side chain of critical active site residue R201 (Figs 3c and 4a). However, differential scanning fluorimetry (DSF) showed no significant difference in the melting temperatures ($T_m$) of LpxBVVL, LpxBVVL-N316A, and LpxBVVL-R201A, and LpxBVVL-F298EN316A had a $T_m$ only 0.5 °C lower than that of LpxBVVL (Supplementary Table 1). Therefore, the

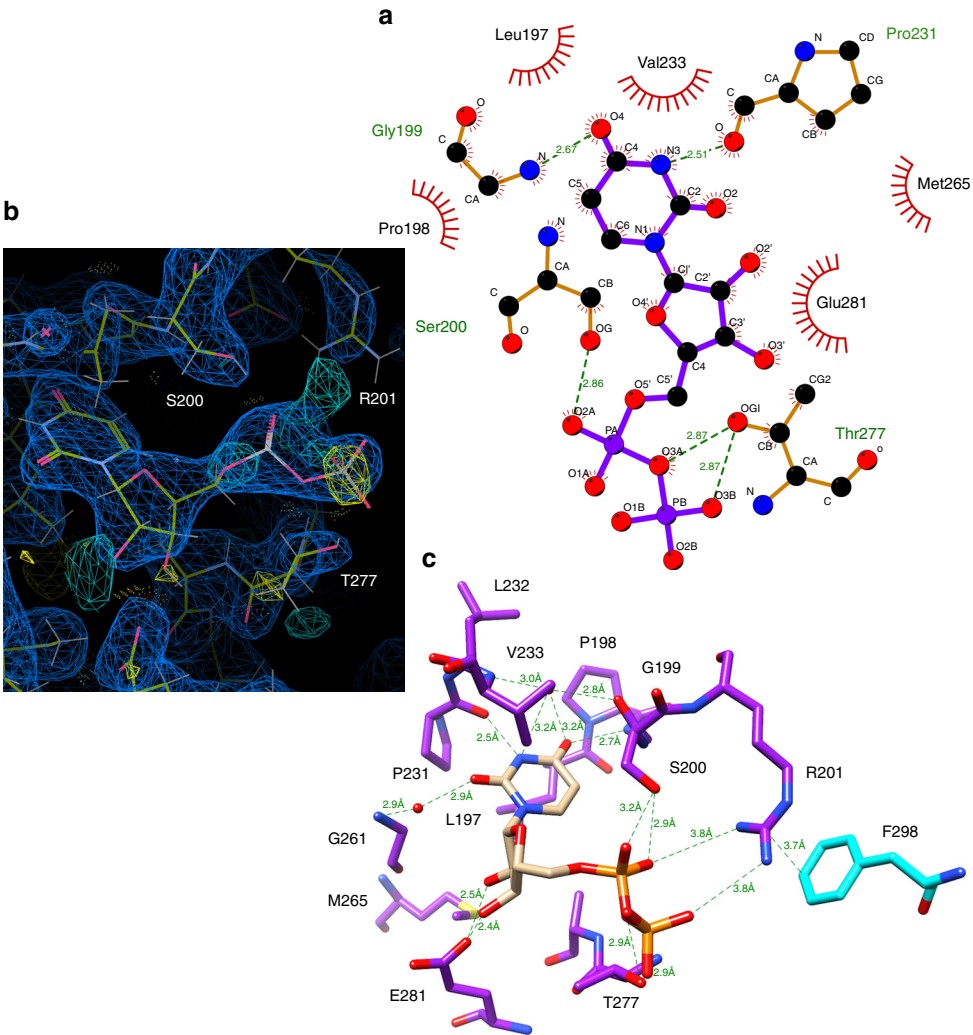

**Fig. 3** LpxB nucleotide-binding pocket. The uracil of UDP binds in a hydrophobic pocket formed by L197, P198, P231, and V233. Putative hydrogen-bond distances in angstroms are as follows: G199 N to uracil O4 (2.7), G199 O to water to uracil O4 (2.8, 3.2), V233 N to water to uracil O4 (3.0, 3.2), P231 O to uracil N3 (2.5), G261 N to water to uracil O2 (2.9, 2.9), E281 Oε2 to ribose O2' (2.5), E281 Oε2 to ribose O3' (2.4), S200 Oγ to α-phosphate O1 (3.2), S200 Oγ to α-phosphate O2 (2.9), T277 Oγ to bridging pyrophosphate O (2.9), and T277 Oγ to β-phosphate O2 (2.9). R201, which is critical for activity[10], is interacting with the α-phosphate O2 and β-phosphate O1 at 3.8, via its η-nitrogens and appears to be positioned to stabilize the UDP leaving-group. **a** Image generated in LigPlot+[69]. **b** Image generated in Coot[62], showing the electron density maps for UDP and the surrounding residues. The 2mFo-DFc map (blue mesh) is shown at 0.4317 e Å$^{-3}$ (0.99 r.m.s.d.), and the mFo-DFc map (cyan mesh for positive and yellow mesh for negative) is shown at 0.45 e Å$^{-3}$ (3.01 r.m.s.d.). See Supplementary Figure 6 for stereo view. **c** Image generated in UCSF Chimera[49]

loss of activity in LpxBN316A and LpxBFN is not simply related to protein unfolding.

The genetic complementation experiments generally support the conclusions of the in vitro assays. The genomic copy of *lpxB* was knocked out with a DNA fragment containing a kanamycin resistance gene flanked by the 3′-end of *lpxB* and the region upstream of the 5′-end in the presence of the wild-type, N316A, or F298EN316A *lpxB* genes expressed from its own operon's promoter[46] on a plasmid. In contrast, the knockout was not obtained when the gene for LpxB6S, LpxB6S-R201A, or LpxBR201A was substituted on the plasmid. These results suggest that the hydrophobic patch (V66, V68, L69, L72, L75, and L76) is essential for productive membrane association or substrate binding. In addition, these results suggest there may be a greater difference between the activity of LpxBFN and LpxBR201A under physiological conditions than what was observed by UDP-release assays, wherein the specific activities were not significantly different (Table 2). These results appear to be consistent with TLC-based assays: LpxBFN produced a detectable amount of lipid

A disaccharide after 3 h (Supplementary Fig. 4), but the lipid A disaccharide band in LpxBR201A reactions only became unambiguous when the reactions were run overnight (Supplementary Fig. 3). Regardless, the cause of this incongruity between the genetic complementation experiments and the specific activities is unclear.

**Oligomerization of LpxB in solution.** Determination of the oligomeric state of LpxB in solution has been hampered by the tendency of LpxB to form large, soluble aggregates. These aggregates are visible on a size-exclusion chromatogram (Supplementary Fig. 1), and this aggregated state was described as an apparent octamer that could be broken into an apparent dimer with *n*-Dodecyl-β-D-Maltopyranoside (DDM) detergent by Metzger and Raetz[10]. The elongated shape of LpxB further hinders the ability of size exclusion to distinguish between the monomer and dimer. In spite of this, analytical ultracentrifugation supported dimerization giving a size between 71 (right wells) and 79 kDa (left wells) (Supplementary Fig. 5). The total dataset

**Table 2 LpxB-specific activities**

| LpxB | Specific activity $\pm$ SE ($\mu mol\ min^{-1}\ \mu mol^{-1}$) | 95% Confidence interval | Percent of wild type |
|---|---|---|---|
| Wild-type | $6.17 \pm 5.3*10^{-1}$ | 4.99 to 7.34 | 100 |
| L72SL75S | $1.08*10^{-1} \pm 2.3*10^{-2}$ | $5.62*10^{-2}$ to $1.59*10^{-1}$ | 1.74 |
| L72SL76 S | $1.99 \pm 3.3*10^{-1}$ | 1.21 to 2.76 | 32.2 |
| L75SL76S | $1.89 \pm 2.3*10^{-1}$ | 1.35 to 2.42 | 30.6 |
| V66SV68SL69S (VVL) | $3.26*10^{-3} \pm 5.5*10^{-4}$ | $2.02*10^{-3}$ to $4.49*10^{-3}$ | $5.28*10^{-2}$ |
| L72SL75SL76S (LLL) | $6.34*10^{-5} \pm 1.04*10^{-5}$ | $4.03*10^{-5}$ to $8.66*10^{-5}$ | $1.03*10^{-3}$ |
| 6S | Below detection limit | ND | ND |
| N316A (NA) | $2.11*10^{-2} \pm 1.5*10^{-3}$ | $1.78*10^{-2}$ to $2.45*10^{-2}$ | $3.42*10^{-1}$ |
| F298AN316A (FN) | $4.80*10^{-4} \pm 6.2*10^{-5}$ | $3.33*10^{-4}$ to $6.28*10^{-4}$ | $7.79*10^{-3}$ |
| R201A (RA) | $3.16*10^{-4} \pm 2.3*10^{-5}$ | $2.66*10^{-4}$ to $3.66*10^{-4}$ | $5.12*10^{-3}$ |
| FN + RA (50% FN) | $1.06*10^{-2} \pm 9*10^{-4}$ | $8.50*10^{-3}$ to $1.27*10^{-2}$ | $1.72*10^{-1}$ |
| FN + RA (37% FN) | $3.63*10^{-3} \pm 3.7*10^{-4}$ | $2.76*10^{-3}$ to $4.50*10^{-3}$ | $5.89*10^{-2}$ |
| FN + RA (24% FN) | $2.91*10^{-3} \pm 1.6*10^{-4}$ | $2.55*10^{-3}$ to $3.27*10^{-3}$ | $4.71*10^{-2}$ |
| FN + RA (15% FN) | $2.35*10^{-3} \pm 2.6*10^{-4}$ | $1.74*10^{-3}$ to $2.95*10^{-3}$ | $3.81*10^{-2}$ |
| FN + RA (7.3% FN) | $1.03*10^{-3} \pm 1.5*10^{-4}$ | $6.70*10^{-4}$ to $1.38*10^{-3}$ | $1.66*10^{-2}$ |
| NA + RA1:1 | $1.06*10^{-2} \pm 1.3*10^{-3}$ | $7.56*10^{-3}$ to $1.35*10^{-2}$ | $1.71*10^{-1}$ |
| Wild-type (NDSB-201) | $3.42 \pm 6.0*10^{-1}$ | 1.994 to 4.84 | 55.4 |
| 6S (NDSB-201) | Below detection limit | ND | ND |

Reactions were performed in triplicate with 1 nM to 10 μM LpxB (as appropriate per variant), 31 μM UDP-DAG, ~ 0.13 mM lipid X, 1 mg ml$^{-1}$ BSA, 0.1 M Tris-HCl pH 8.0, and 0.1% Triton X-100 at ambient temperature (21 °C). Reactions were quenched and UDP was quantified using a UDP-Glo Glycosyltransferase Assay kit (Promega), which quantifies free UDP by a luciferase coupled reaction. Specific activities with standard errors and 95% confidence intervals were calculated by linear regression of UDP concentrations at three or four time points for each LpxB variant in Graphpad Prism v7.03 ND, insufficient data

did not converge, but the $\chi^2$-residuals were minimized when the highest speed right well was excluded giving a mass of 78.9 kDa, which is most consistent with the expected 84.5 kDa dimer.

To functionally validate the C-terminal swap model, 6His-LpxBR201A was co-expressed with LpxBN316A or LpxBFN, which carry mutations in the C-terminus. We hypothesized that the formation of an intact active site via C-terminal swap of the R201A mutant with one of these other mutants would result in increased activity in the co-purified protein if the second mutant were much less active than the wild type and if the expression levels of the mutants were approximately equal. The ability of His-tagged LpxBR201A to pull down more active LpxBN316A in the presence of 0.2–2% Triton X-100 supports the formation of a stable oligomer (Supplementary Fig. 4). In addition, we mixed purified LpxBR201A with purified LpxBN316A or LpxBFN. However, combining 50% LpxBR201A with 50% LpxBN316A decreased the specific activity by approximately half indicating that the extent of heterodimer formation was not sufficient to overcome the initial ~ 50% dilution in activity (Table 2). As these LpxB mutants lost activity when left overnight at 4 °C (data not shown), dimerization could not be allowed to reach equilibrium; therefore, the change in activity depends on the kinetics of subunit exchange as well as the difference in activity of mutant and intact active sites. Hence, the failure of 50% LpxBN316A with 50% LpxBR201A to show increased activity over 100% LpxBN316A likely reflects slow dimer exchange since the activity of LpxBN316A was measured well below 25% of wild type (Table 2). In order to obtain a sufficiently impaired mutant to observe an increase in activity upon sub-equilibrium dimer exchange, a second mutation was made in the swapped C-terminus (F298E) producing LpxBFN as discussed above. The F298E mutation was selected based on the hypothesis that R201 stabilizes the negative charge of the UDP leaving group during catalysis[10]. As F298 already forms a cation-π interaction with R201, E298 is hypothesized to form a salt-bridge with R201 thereby balancing this critical positive charge and altering the conformation of R201 from that required for catalysis. When 50% LpxBR201A was combined with 50% LpxBFN, the sample showed significantly more activity than freshly thawed 100% LpxBFN (Table 2). Increased activity was also observed with 63,

76, 85, and 92.7% LpxBR201A, and this activity decreased with increasing LpxBR201A concentration (Table 2). These data support the C-terminal swap model because the C-terminal swap would form one intact active site per R201A-F298EN316A dimer. A R201A-F298EN316A heterodimer with one intact active site would be expected to produce more activity than an LpxBFN homodimer because LpxBFN has only 0.0078% of wild-type activity (Table 2). Since minimal loss of protein stability was associated with the F298EN316A mutations (Supplementary Table 1), this increase in activity is consistent with the formation of C-terminally swapped heterodimers.

## Discussion

The tertiary structure of LpxB is similar to many previously characterized GT-B enzymes[23,38,43,45,47,48]; however, LpxB has a unique quaternary structure wherein the protein fold is completed by the last 87 residues of the opposite subunit in the dimer. This C-terminal swap has not been observed before in this protein superfamily, but a sialyltransferase from this superfamily involved in LPS synthesis in *Neisseria meningitidis* shows a distinct N-terminal swap of the first 130 residues[45]. These unique domain arrangements likely contribute to the stability of the enzyme dimer and may potentially help coordinate activities between the two molecules.

LpxB utilizes similar active site residues as previously characterized GT-B enzymes (Table 3), in particular MurG[38] (18% overall sequence identity) (Supplementary Fig. 6). The most conserved binding motif is the use of Glu (LpxB E281) to hydrogen-bond with the 2′- and 3′-hydroxyls of the ribose of the sugar donor substrate. In addition, S17, which was shown to be important for LpxB activity[10], is positioned similarly to the critical residue T16 in MurG[38]. Finally, although the predicted catalytic base of LpxB (D98)[10] is not close in primary sequence to that of MurG (H19)[38], the side chains of these residues are close in space when the structures are overlaid[49] (Fig. 5a).

However, there are also important differences between the sugar donor binding site of LpxB and MurG. LpxB T277 appears to correspond to T266 in MurG, but T277 hydrogen-bonds with the β-phosphate, whereas T266 hydrogen-bonds with the α-phosphate[38]. Conversely, MurG S192 and LpxB S200 hydrogen-

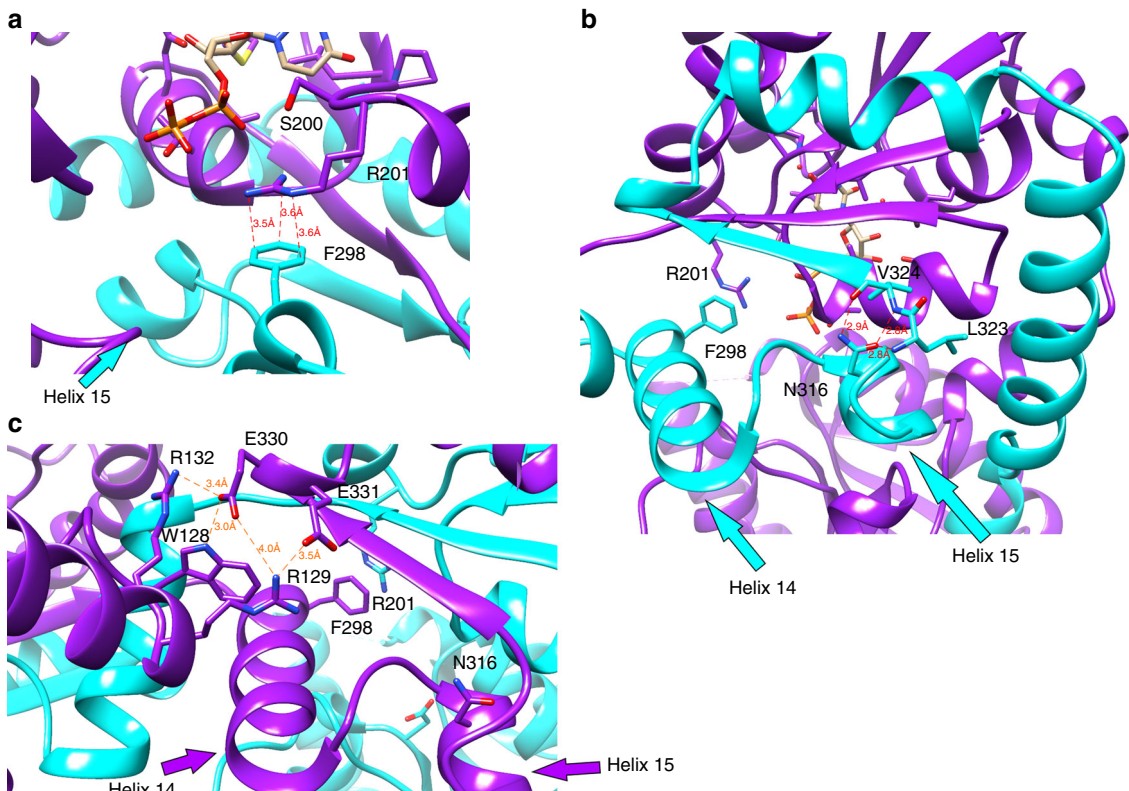

**Fig. 4** Interactions that stabilize the dimeric structure. **a** Position of F298 in relation to the critical residue R201. These residues appear to be engaged in a cation–π stacking interaction with a distance of 3.6 Å between ηN1 and ζC, a distance of 3.5 Å between ηN2 and εC1, and a distance of 3.6 Å between εN and δC2. However, the R201 side chain is not fully resolved in the electron density (Fig. 2). The position of helix 15, which contains N316 is also visible. **b** Position of N316 relative to the crossover helix (helix 14), showing how N316 appears to stabilize the turn from helix 15 to the final strand of the C-terminal β-sheet (strand 13). N316 hydrogen-bonds with the backbone of residues 323 and 324. The interaction distances in angstroms are as follows: N316 δO to L323 N (2.8), N316 δO to V324 N (2.8), and N316 δN to V324 O (2.9). **c** Interactions between the N-terminal domain and the adjacent C-terminal domain. R129 forms a salt-bridge with E331 (3.5 Å between ηN1 and εO2). E330 forms a salt-bridge with R132 (3.4 Å between εO1 and ηN2) and a hydrogen-bond with W128 (3.0 Å between εO1 and εN)

bond with the β- and α-phosphates, respectively. MurG S192 is part of a conserved GGS loop[38] while S200 is part of a conserved PGSR loop that also includes the essential residue R201[10]. Binding of the uracil moiety similarly shows variations between the enzymes: LpxB G261 is positioned similarly to MurG I245, which hydrogen bonds uracil N3 and O4 via its backbone amide nitrogen and carbonyl[38], but the amide of LpxB G261 hydrogen bonds to uracil O2 via an intervening water. In LpxB, uracil N3 and O4 are hydrogen-bound by the backbone amides of P231, G199, and a water bound to the V233 and G199 amides. In MurG, the uracil O2 is bound by R164, a residue in the first linker between the N- and C-terminal domains[38]. Finally, MurG Q288, Q289, and N128 contact the 3′- and 4′-hydroxyls of GlcNAc; however, the corresponding helix is involved in the beginning of the C-terminal swap in LpxB and is thus not positioned to contact a glucosamine bound to UDP. These structural differences suggest that LpxB also has a distinct mode of binding for the glucosamine moiety of the natural substrate, UDP-DAG. The β-hydroxy-myristoyl moieties on the 2′-amine and 3′-hydroxyl of the glucosamine in UDP-DAG, likely have key roles in substrate binding. This may explain some of the differences in binding modes of the LpxB and MurG structures.

LpxB in our crystal structure appears to be in an open conformation. As observed for MurG[38,39], PimA[48,50], and other GT-B enzymes[40], the active site cleft of LpxB probably closes during productive substrate binding to bring the substrates into the correct conformation for nucleophilic attack by the lipid X 6′-hydroxyl on the anomeric carbon of UDP-DAG. In MurG, this involves a ~ 10° change in the relative positions of the globular domains with the linker region acting as the hinge[38]. Although the intertwined dimerization mode may limit the movement of the C-terminal domain of LpxB, the contacts between the N-terminal domain and the adjacent C-terminal domain are more limited, consisting of salt bridges between R129 and E331 and between R132 and E330 and a hydrogen-bond between W128 and E330 (Fig. 4c). Therefore, a similar hinge-like movement of the N-terminal domain may be possible. A large conformational change associated with hydrolytic removal of UMP from UDP-DAG was observed for LpxI, an enzyme that generates lipid X in a subset of Gram-negative bacteria[25,26]. Thus, further studies are needed to capture different structural states of LpxB to fully understand its catalytic mechanism.

Previous structural studies of *E. coli* MurG revealed the presence of a surface-exposed hydrophobic patch (I75, L79, F82, W85, and W116) surrounded by a basic horseshoe (K69, K72, R80, R86, R89, and K140)[39]. As mentioned above, the comparison of MurG to a homology model[37] of LpxB enabled the prediction of the LpxB hydrophobic patch. These residues form a similar hydrophobic and basic face in the two proteins; however, the exact positions of the residues is not well conserved (Fig. 5c). PimA, which is a retaining glycosyltransferase that transfers mannose to phosphatidylinositol (16% identity to LpxB), also has a similar hydrophobic and basic face[48] (Fig. 5c). One particularly similar feature is the flexible loop containing predicted

**Table 3 Comparison of GT-B active sites**

| Enzyme | PDB | Inverting/ retaining | Catalytic base | | | | | Pyrophosphate | | Pyro phos. | Ribose |
|---|---|---|---|---|---|---|---|---|---|---|---|
| LpxB | 5W8X | I | E15 | S17 | D98 | P198 | G199 | S200 | R201 | T277 | E281 |
| MurG | 1NLM | I | NA | T16 | H19 | G190 | G191 | S192 | NA | T266 | E269 |
| Δ24PmST1 | 2IHZ | I | NA | S36 | D141 | NA | NA | NA | NA | NA | E338 |
| MshA | 3C4V | R | NA | NA | NA | NA | NA | NA | NA | NA | E324 |
| NST | 2YK7 | I | NA | NA | D258 | NA | NA | S322 | NA | NA | E300 |
| PimA | 2GEJ | R | Y9 | NA | NA | NA | NA | NA | R196 | NA | E282 |
| WaaA | 2XCU | I | NA | NA | E31 | NA | NA | NA | R212 | NA | E276 |

Residues shown are those found to be analogous to LpxB active site residues by comparison of listed crystal structures
NA, not applicable, indicates that an analogous residue could not be unambiguously assigned

membrane-binding hydrophobic residues followed by an amphipathic helix with a highly basic N-terminal end (Fig. 5b). In PimA, the flexible loop includes residues 59–70, and the following surface-exposed basic residues are H76, R77, K78, K80, and K81[48]. In LpxB, the disordered loop is residues 62–71 and the basic residues are R73, R74, H77, K84, and R85 (Supplementary Fig. 7). Deletion of the loop or mutation of the basic residues to Ser prevented catalysis by PimA[48]. Likewise, mutation of hydrophobic residues in this region of LpxB decreased the activity of LpxB. In PimA, the relevance of this helix to membrane association was further demonstrated by Förster resonance energy transfer between W82 in the helix and fluorescently labeled lipids in small unilamellar vesicles and by blue-shifting of the fluorescence emission of W82 upon association with vesicles[51]. These data showed that the amphipathic helix associates with vesicles containing negatively charged lipids, in particular the acceptor substrate, phosphatidylinositol[51]. The membrane-association role of the amphipathic helix is likely conserved in LpxB[40]. Although we found LpxB6S to be inactive toward lipids in mixed micelles, the soluble enzyme remained catalytically competent when the substrates were solubilized with non-detergent sulfobetaine 201. In addition, the mutated versions of PimA and LpxB were still able to bind the nucleotide of their sugar-donor substrates[48]. Therefore, the data support the hypothesis that membrane surface-active GT-B enzymes require surface-exposed hydrophobic and basic residues to extract their lipid substrates from the membrane[40,51].

The distance (~ 15 Å) of the predicted catalytic base (D98) from the putative membrane-binding region (loop 62–71 and helix 4) suggests that LpxB mostly or fully extracts its lipid substrates from the membrane. The preceding enzyme, LpxH, and its alternative, LpxI, were observed to fully bind their substrates[20,21,25]. In contrast, the following enzyme in the pathway, LpxK, was observed to primarily bind lipid head groups with the hydrocarbon tails disordered and extending into solvent[52]. Regardless, LpxB, LpxH, and LpxK do not exhibit strong selectivity for chain length[19,53]. LpxB can even utilize lipid X derivatives with a single chain or a third chain attached to the hydroxyl of the 2′-β-hydroxymyristate, albeit with two to three orders of magnitude less activity than for lipid X[54,55]. Thus, the presence of the acyl chains may help the substrates orient productively in the active site of LpxB, but the substrates probably are not subjected to molecular ruler binding as observed in LpxA[17] and LpxD[19].

Molecular docking[56] was utilized to generate plausible models for the binding of the natural donor substrate of LpxB, UDP-DAG (Fig. 6). The models are largely consistent with the observed position of UDP in the crystal structure (Fig. 6). In both models shown, the lipid tails extend toward a hydrophobic groove lined by V125, W126, and W128; in addition, the anomeric carbon is exposed for nucleophilic attack from above and towards the N-terminal domain in both models. This suggests that lipid X binds second and on top of UDP-DAG, possibly with its lipid tails

extending farther along the hydrophobic groove, which would explain why we were unable to obtain a plausible docking model for lipid X binding. However, the top ranked model (1 of 9) (Fig. 6a) places the anomeric carbon 8.8 Å from the predicted catalytic base (D98). A hinge-like movement of the N-terminal domain might position D98 to accept a proton from the nucleophilic 6′-hydroxyl of lipid X. On the other hand, a lower ranked model (7 of 9) (Fig. 6b) places the anomeric carbon 5.4 Å from D98, which should allow D98 to accept a proton from the attacking hydroxyl with minimal movement of the domain.

In summary, we have solved the structure of LpxB, revealing a new twist on the conserved GT-B fold wherein LpxB forms a dimer with the C-terminus of one subunit completing the fold of the second subunit. Soaking with UDP-GlcNAc revealed the location of the nucleotide-binding region of the active site and explained the role of previously identified conserved residues that are important for activity[10]. Furthermore, our activity assays of solubilized forms of LpxB helped to provide more insight into the mode of membrane association of surface-active membrane proteins such as LpxB, PimA, and MurG. This structure should aid in the rational and computational development of new antibiotics targeting LpxB to combat the increasing problem of antibiotic resistance.

## Methods

**Cloning and purification of LpxB.** The *lpxB* coding sequence was amplified from *E. coli* BL21 cells with Phusion DNA polymerase (NEB) using the forward and reverse primers carrying BsaI (NEB) and XbaI (NEB) restriction sites, respectively. The *lpxB* coding sequence was inserted into pE-SUMO expression vector (Life-Sensors) to attach an N-terminally His-tagged SUMO to the N-terminus of LpxB. Following purification, the His-tagged SUMO was removed by proteolytic cleavage with the SUMO protease Ulp1. Alternatively for co-expression and purification of two mutants of LpxB, *lpxB* genes were inserted into pRSF-Duet-1 (Novagen) at the EcoRI (TaKaRa) and HindIII (NEB) sites of MCS1 and at the NdeI (NEB) and KpnI (NEB) sites of MCS2, thus encoding one N-terminally His-tagged protein and one untagged protein.

BL21 (DE3) cells (Lucigen) were used to express LpxB: transformed cells were grown in Miller's Luria Broth (RPI) to an OD$_{600}$ 0.5–0.9 at 37 °C and then were cooled to 18 °C and induced with 1 mM isopropyl-β-D-1-thiogalactopyranoside (IPTG) (GoldBio) overnight. For producing selenomethionine derivative proteins, the cells were grown in M9 media, and methionine synthesis was inhibited by addition of an amino acid cocktail in addition to 50–75 mg l$^{-1}$ selenomethionine (Chem-Impex Int'l) 15 min before induction with IPTG[57]. Cells were pelleted at 4,540 × g (4 °C) for 25 min (Beckman J6-MI, JS-4.2) and then resuspended in lysis buffer (0.5 M NaCl (Fisher), 50 mM Tris-HCl (Fisher) pH 7.4, 10% glycerol (v/v) (Fisher), 5–10 mM β-mercaptoethanol (EMD Millipore)). Cells were lysed via sonication with a Branson Sonifier 450 (50% duty cycle, output 5, 6 min in three 2 min internals) while on ice. For cells expressing SUMOylated wild-type LpxB, 1% DDM (w/v) (Anatrace D310) was added to lysate and mixed for 1 h (4 °C) to solubilize SUMO-LpxB. Lysate was centrifuged at 63,988 × g (4 °C) for 45 min (Beckman Avanti J-25 I, JA-25.50) to remove cell debris. Supernatant was batch bound for 1 h (4 °C) to HisPur Ni-NTA resin (Thermo) from 4 to 5 ml of slurry that had been equilibrated with 20 ml lysis buffer with 10 mM imidazole (Chem-Impex Int'l). Resin was collected with a gravity-flow column (room temperature) and washed with ice cold lysis buffer with 25 mM (50 ml) and 40 mM (15 ml) imidazole. SUMO-LpxB was eluted from the column with 15 ml lysis buffer with 250 mM imidazole. After addition of Ulp1, the elutant was dialyzed against 500 ml storage buffer (0.3 M NaCl, 20 mM Tris-HCl pH 7.4, 5% glycerol, 5–10 mM

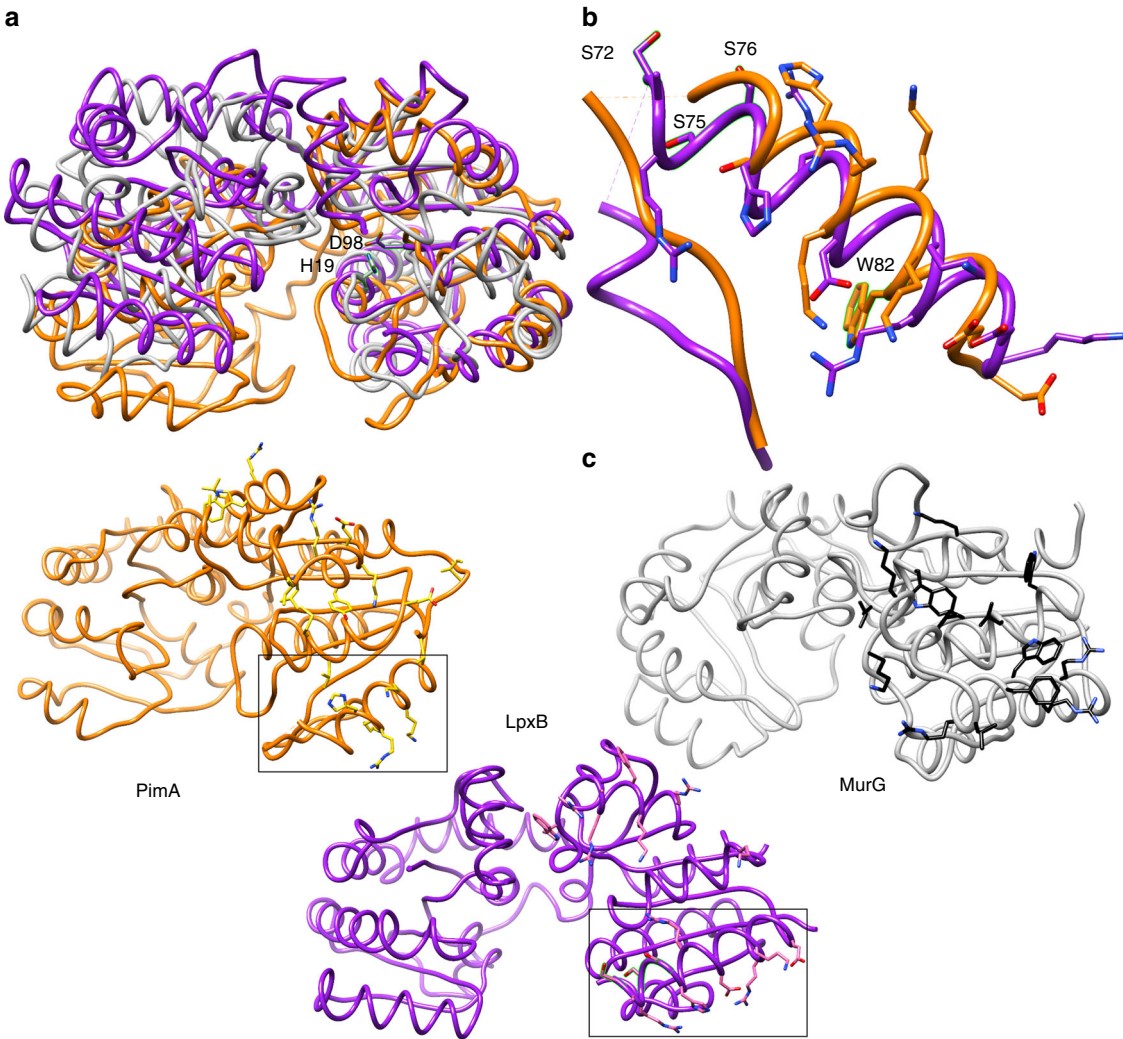

**Fig. 5** Membrane association. **a** Overlay of the N-terminal domains of LpxB7S (purple) with MurG (grey) (PDB: 1F0K)[39] and PimA (orange) (PDB:2GEJ)[48]. The catalytic bases of LpxB and MurG, D98 and H19, respectively, are also shown. **b** Region of the LpxB-PimA membrane-binding face of particular interest. Deletion of the hydrophobic, flexible loop or mutation of basic residues in this helix to Ser inactivated PimA[48], and mutation of hydrophobic residues in this loop and helix to Ser (visible residues highlighted) decreased LpxB activity. In addition, W82 in the PimA helix was shown to interact with negatively charged lipid membranes[51]. **c** Side-by-side comparison of the 3 proteins, showing residues that may be important for defining the membrane association surface. While the specific residues are not well conserved, they appear to form a similar hydrophobic and basic surface in all three enzymes. The regions of LpxB and PimA shown in **b** are boxed

dithiothreitol (GoldBio), (0.05% DDM for wild-type LpxB)) overnight (4 °C) in 3.5 kDa cutoff tubing (Spectra/Por). Cleaved SUMO-LpxB was run back through the Ni-NTA column to remove His-tagged SUMO.

LpxB was concentrated with a 30 kDa cutoff centrifugal filter (Amicon). LpxB was then further purified via size exclusion chromatography on a Superdex 200 column (GE Healthcare) equilibrated in storage buffer. Poorly expressing wild-type LpxB was run through a 10/300 GL column. Mutant forms of LpxB with improved expression (described in "Crystallization of LpxB") were run through a HiLoad 26/60 prep grade column. One or 2 ml fractions were collected, respectively. LpxB fractions were concentrated to 10 mg ml$^{-1}$, as measured by $A_{280}$ ($\varepsilon = 32{,}805$ M$^{-1}$ cm$^{-1}$) (Nanodrop 8000 (Thermo)), for robotic crystal screening.

**Protein mass spectrometry**. The identity of purified LpxB was confirmed via liquid chromatography tandem mass spectrometry. Samples of soluble LpxB (LpxB6S described in "Crystallization of LpxB") were submitted to Novatia (Newtown, PA). Mass/charge peaks for the expected LpxB6S molecular weight and what appears to be a cysteinylated form of LpxB6S were observed (Supplementary Fig. 8).

**Crystallization of LpxB**. Extensive sitting drop (200 nl) robotic screening (Rigaku) of wild-type LpxB revealed no successful crystallization conditions. Thus, we sought to form a crystallizable mutant of LpxB. We reasoned that the DDM micelles required to solubilize wild-type LpxB may interfere with crystal packing, so we

sought to create a soluble form of LpxB. A homology model of LpxB was produced via the Phyre2 server using UDP-N-acetylglucosamine 2-epimerase from *T. thermophilus* HB8 (PDB: 1V4V) for sequence threading[37]. Comparison of the LpxB model with the crystal structures of MurG (PDB: 1F0K and 1NLM)[38,39], another inverting glycosyltransferase (18% identity), allowed us to identify a putative surface-exposed hydrophobic patch (V66, V68, L69, L72, L75, and L76).

Each of these residues was mutated to Ser using the Stratagene QuickChange protocol (Pfu Turbo polymerase (Agilent), DpnI (NEB)), and the expression and behavior on a size exclusion column of various combinations of these mutations was tested (Supplementary Fig. 1). The fully mutated form of the enzyme (LpxB6S) appeared least prone to aggregation and was selected for further robotic crystal screening. Bipyramidal and hexagonal rod-shaped LpxB6S crystals grew in several conditions. The best condition (0.8 M LiCl, 32% PEG 4000, 0.1 M Tris pH 8.5) (MCSG-1 C9 Microlytic) was optimized in 2 µl hanging drops over 500 µl well solution with 1:1 8 mg ml$^{-1}$ LpxB6S and well solution. The optimized condition for LpxB6S crystallization was 0.6–0.8 M LiCl (MP Biomedicals), 35–40% PEG 4000 (Hampton Research, HR2–529), 0.1 M Tris-HCl pH 8.6. This condition led to the formation of both crystal forms in the same drop. Unfortunately, the bipyramidal crystals that diffracted to higher resolution were much rarer than the hexagonal rods. Additive screening (HR2-428 Hampton Research) showed that 10 mM trimethyl-ammonium chloride in the hanging drop, but not in the well solution, selected for the formation of bipyramidal crystals.

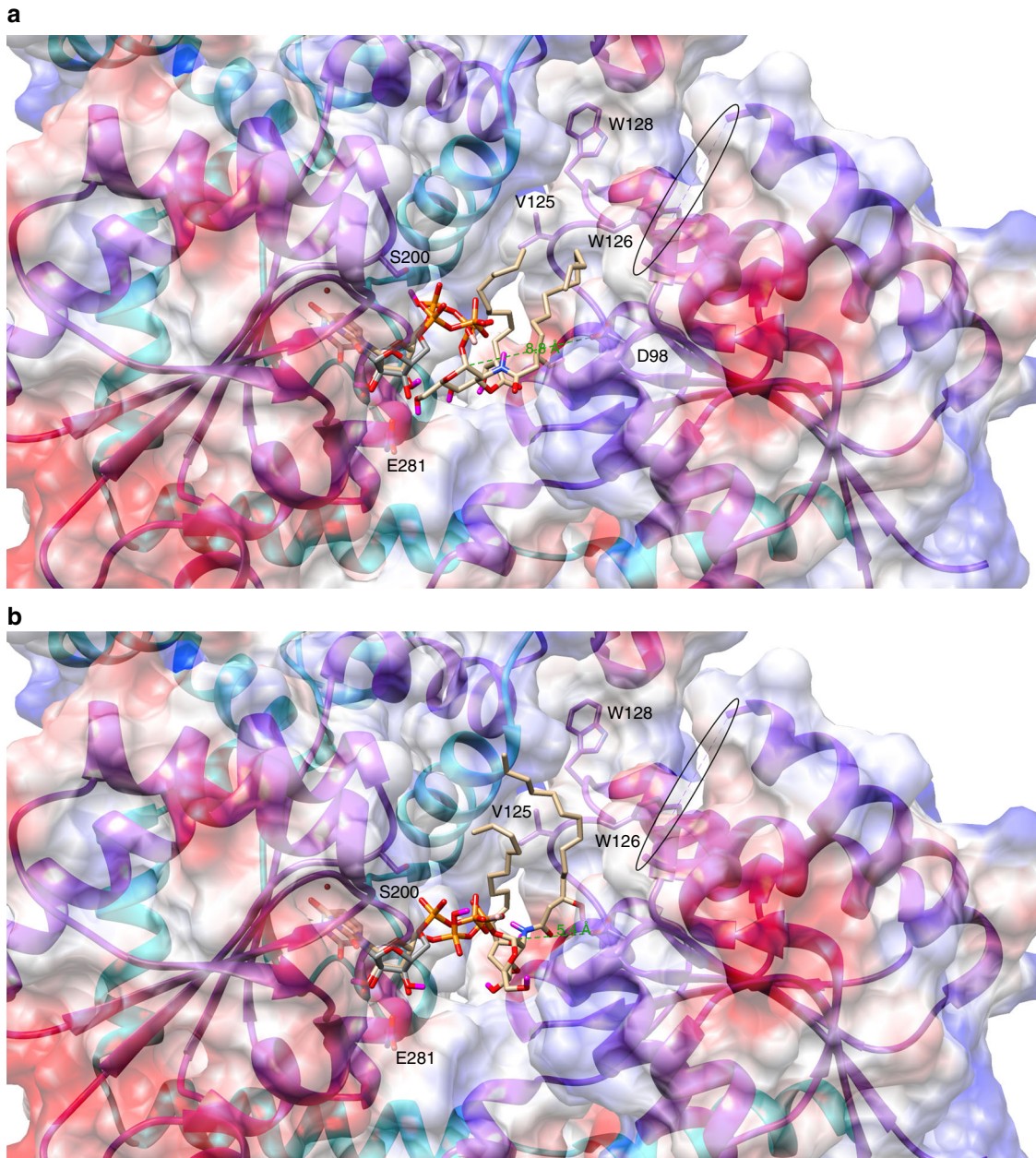

**Fig. 6** Molecular docking model of UDP-DAG binding. UDP-DAG (tan carbons) was docked into the LpxB active site cleft with AutoDock Vina[56]. The software correctly identified the nucleotide-binding site overlaying reasonably well with the position of UDP (gray carbons) observed in the crystal structure. The lipid tails were predicted to extend toward a hydrophobic groove lined by V125, W126, and W128. The break in the chain corresponding to the disordered loop is circled. **a** The top ranked model places the anomeric carbon 8.8 Å from the predicted catalytic base, D98. **b** A lower ranked model (seven of nine) places the anomeric carbon 5.4 Å from D98, which would require less movement of the protein in order to correctly orient D98 and the 6′-hydroxyl of lipid X for proton transfer and nucleophilic attack on the anomeric carbon

Although native LpxB6S bipyramidal crystals diffracted beyond 2.4 Å (Table 1), selenomethionine (Sem) derivative LpxB6S crystals that diffracted well enough for phasing by single-wavelength anomalous diffraction could not be grown. *E. coli* LpxB has 13 Met, more than adequate for phasing a 42 kDa protein. We hypothesized that one or more of these methionines was interfering with crystallization in the Sem derivative, possibly by being surface exposed and decreasing protein solubility. Thus, we further utilized the Phyre2 homology model of LpxB to predict possible culprits. Three Met residues were identified and mutated to Ser; however, only M207S preserved the stability of LpxB6S as judged by size exclusion chromatography (Supplementary Fig. 1). M46S and M295S increased protein aggregation. Thus, the Sem derivative of LpxB6SM207S (LpxB7S) was selected for further crystal screening. Hexagonal rod crystals of the LpxB7S Sem derivative that diffracted to 3.43 Å (Table 1) were obtained in a 2 μl drop of 8 mg ml⁻¹ LpxB7S (0.9 μl), well solution (0.9 μl), and 0.2 μl of 100 mM trimethyl-ammonium chloride (Acros Organics) hanging over 500 μl 28% PEG 4000, 0.35 M

LiBr (Sigma-Aldrich), and 0.1 M Tris-HCl pH 8.6. Switching from LiCl to LiBr was essential for obtaining harvestable LpxB7S Sem derivative crystals.

**Diffraction data collection and model building.** All data sets used for model building and refinement were collected at the Advanced Photon Source at Argonne National Lab on beamlines 24-ID-C and 24-ID-E at 100 K and 0.979 Å. The data were indexed, integrated, and scaled by the Rapid Automated Processing of Data beamline software (XDS[58] and CCP4[59]) or in HKL2000[60] (Table 1). The PHENIX program suite was utilized for initial phasing, model building, and refinement[61]. Manual model building and further refinement were performed in Coot[62] and PHENIX Refine.

The selenomethionine anomalous diffraction data were used to calculate the initial phases and to build a crude model: AutoSol found 11 Se sites with a figure of merit of 0.348 in space group P6₄ and built 201 residues in 16 fragments with an R-

work of 0.3786 and an R-free of 0.4692. Once the R-free of this model dropped below 0.4, this model was used as a molecular replacement model to phase the higher resolution native data in PHENIX Phaser. Sixteen fragments totaling 324 residues were built automatically in PHENIX AutoBuild in space group P3$_2$21 with an R-Work of 0.2930 and an R-Free of 0.3775. The rest of the structure was built manually in Coot with iterative refinement in PHENIX Refine. In the higher resolution crystal form, there is one LpxB molecule in the asymmetric unit. Three structures were refined. The apo LpxB6S (PDB: 5W8N) structure was refined with Ramachandran statistics of 99.18% favored, 0.55% allowed, and 0.27% outliers. The apo LpxB7S structure (PDB: 5W8S) was refined with 98.63% favored, 1.10% allowed, and 0.27% outliers. The UDP-bound LpxB7S structure (PBD: 5W8X) was refined with 98.34% favored, 1.10% allowed, and 0.55% outliers.

**Analytical ultracentrifugation**. Samples of LpxB6S were submitted to the Spectroscopy and Biophysics Core at University of Nebraska-Lincoln for analytical ultracentrifugation. LpxB6S (120 μl at 1.0, 0.5, and 0.25 mg ml$^{-1}$) was centrifuged at 5,000, 7,000, 9,000, 12,000, or 16,000 r.p.m. (2,016, 3,951, 6,532, 11,612, or 20,644 × g) for 10 h in a ProteomeLab XL-A ultracentrifuge (Beckman) and absorbance was measured at 280 nm. Only the 1 mg ml$^{-1}$ protein produced usable data. The experiment was run in triplicate, but the middle wells contained an absorbance spike, and these wells were excluded from modeling. A solvent density of 1.0 g ml$^{-1}$ and a protein partial specific volume of 0.73 ml g$^{-1}$ were utilized for modeling the protein mass.

**Enzymatic assays**. Genetic knockout/complementation assays of LpxB were performed in the BW25113/pKD46 strain of *E. coli* containing Lambda Red recombinase under L-arabinose induction. The cells were made chemically competent[63] and transformed with a pACYC-Duet plasmid (Novagen) containing the genes for wild-type *lpxB*, *lpxBR201A*, *lpxBN316A*, *lpxBF298EN316A*, *lpxB6S*, or *lpxB6SR201A*. The plasmids contained the wild-type promoter for *lpxB* operon (104 bp upstream of the *fabZ* start codon) (Operon Database v3[46]) ligated into the *Bam*HI (NEB) and *Eco*RI sites and the *lpxB* genes ligated into the *Eco*RI and *Hin*dIII sites of MCS1. For experiments testing the ability of different mutants to complement each other by C-terminal swapped dimerization, a second *lpxB* gene was ligated into the *Nde*I and *Kpn*I sites of MCS2. These strains were then grown in the presence of 1 mM L-arabinose (GeneMate), 100 mg/L ampicillin (Cayman Chemical Company), and 30 mg l$^{-1}$ chloramphenicol (Sigma), made chemically competent[63], and transformed with a DNA fragment containing a kanamycin resistance gene flanked by 50 bp upstream of *lpxB* and the last 24 bp of *lpxB* followed by 51 more basepairs downstream. Chloramphenicol (30 mg l$^{-1}$), carbenicilin (GoldBio) (100 mg l$^{-1}$), and L-arabinose (1 mM) were also included in the growth step of transformation, which was extended to 3 h. Knockout strains were selected on Miller LB agar (BD Difco) plates with 40 mg l$^{-1}$ kanamycin (Teknova).

In vitro enzymatic assays were performed with various soluble and membrane-associated mutants of LpxB and with lipid X and UDP-DAG purified from Ni-NTA purified wild-type and D225A CcLpxI[25], respectively. For enzymatic assays, insoluble versions of LpxB were purified using Triton X-100 (Acros) instead of DDM (2% for solubilization, 0.2% for Ni-affinity column, and 0.1% for size exclusion). LpxI was expressed in C41 DE3 *E. coli* (Lucigen) and purified as for soluble LpxB except that the LpxI lysis buffer was 50 mM NaCl and 20 mM HEPES-NaOH pH 8.0. Eluted protein was then concentrated in 10 kDa cutoff centrifugal filters (Amicon). Wild-type LpxI was dialyzed against 500 ml LpxI lysis buffer, and LpxID225A was dialyzed against 50 mM NaCl and 20 mM Bis-Tris (Sigma) pH 6.0. LpxI precipitated during dialysis, but the substrates remained bound. Two volumes of methanol (Fisher) were added to the dialyzed protein, which was vortexed and then centrifuged at 3,000 × g (4 °C) for 30 min (Eppendorf 5702 R) to remove precipitate. For UDP-DAG the supernatant was concentrated in a Savant ISS110 SpeedVac (Thermo) set on medium (43 °C), and no further purification was performed. For lipid X, the supernatant was adjusted to pH 2 with HCl (Fisher) and dichloromethane (Sigma-Aldrich) (volume equal to methanol volume) was combined with the supernatant in a separatory funnel. Lipid X was extracted into the lower (CH$_2$Cl$_2$) phase, which was collected and then evaporated under vacuum. Lipid X residue was suspended in LpxI lysis buffer. The identities of the purified lipids were confirmed by mass spectrometry with an Agilent 1100 LC/MSD TOF mass spectrometer (G1969A) (Supplementary Fig. 9).

For TLC-based assays, substrates and enzyme were diluted into 2 × reaction buffer (40 mM Tris-HCl pH 7.8, 0.1% (w/v) Triton X-100, and 1 mg ml$^{-1}$ bovine serum albumin (BSA) (Sigma)) unless otherwise noted, and 10 μl reactions were run at 30 °C (Bio-Rad DNAEngine). UDP-DAG was used at 0.11 mM (as measured by UDP A$_{260}$ (Nanodrop 8000) with ε ~ 9.9 mM$^{-1}$ cm$^{-1}$), and lipid X was used at ~ 0.13 mM as determined by lipid X depletion after reaction with 0.13 mM UDP-DAG. LpxB was used at 0.5 mg ml$^{-1}$, and the final pH was adjusted to 8 with 100 mM Tris-HCl pH 8.0. Reactions were quenched by spotting on HPTLC Silica gel 60 plates (EMD Millipore). Once dry (~ 1 h after spotting), TLC plates were run with 25/15/4/2 chloroform (Macron)/methanol/water/acetic acid (Fisher) (v/v/v/v). TLC plates were then allowed to dry at least one hour before they were sprayed with 20% sulfuric acid (Sigma-Aldrich) in ethanol (Decon) and charred with a Dual-Temp Heat Gun (Genesis) set on high (538 °C). The reactants and lipid A disaccharide product appeared as darker bands. In addition, fluorescent (F$_{254}$) TLC

Silica gel 60 plates (EMD Millipore) allowed the visualization of UDP-DAG as a fluorescent shadow by UV transmission (Bio-Rad Gel Doc EZ Imager) before charring.

Semi-quantitative comparisons of the relative activities of LpxB mutants were generated by performing the TLC analysis for reactions run for various times. Reactions were judged to have reached completion when the product was visible and the limiting reagent (UDP-DAG) was no longer visible. The maximum time when the reactions were not complete and the minimum time when the reactions were complete are presented as ranges (Supplementary Table 2). In addition, the amount of the lipids present on each plate could be compared by analyzing band intensity in ImageJ[64]. TLC plate charring is an established methodology for lipid quantification though absolute quantification requires standard curves of the lipids on interest to be run on the same plate as the samples[65,66]. The positive correlation between lipid amount and band intensity was confirmed for this study by running different amounts of UDP-DAG and lipid X and by reacting different amount of these reactants to completion with wild-type LpxB (Supplementary Fig. 10). The correlation was found to be positive and linear (Supplementary Fig. 10C and F).

Although TLC-based assays have the benefit of visualizing the product of interest, they require standard curves for quantification. Thus, to produce more quantitative comparisons of LpxB activity, UDP release was quantified using a UDP-Glo Glycosyltransferase Assay kit (Promega) as described by the manufacture. A twofold UDP (Promega) dilution series was prepared in 0.1 M Tris-HCl pH 8.0, 0.1% (w/v) Triton X-100, and 1 mg ml$^{-1}$ BSA. Reactions (10 μl) were run under these buffering conditions with 31 μM UDP-DAG, ~0.13 mM lipid X, and 1 nM to 10 μM LpxB at ambient temperature (~ 21 °C) in a white 384-well plate (Greiner Bio-One 781,074). LpxB reactions were quenched with 10 μl UDP Detection Reagent (Promega) and the plate was shaken 30 s at 1,440 r.p.m. (1 mm amplitude) with a Spark 10 M plate reader (Tecan). The luciferase coupled reaction was placed at ambient temperature for 1 h. The resulting luminescence was then measured at 555 nm with the Spark 10 M using the TR-FRET setting: 900 nm excitation, 100 μs lag time between excitation and reading, 2 ms integration of 555 nm emission, and 200 reads averaged together. Luminescence was converted to UDP concentration with concurrently run UDP standard curves prepared with 10 μl aliquots from the UDP dilution series and 10 μl UDP Detection Reagent. Reactions without LpxB served as a negative control: UDP detected at or below the concentration in these samples was assumed to result from UDP-DAG hydrolysis rather than formation of lipid A disaccharide. Of note, the UDP concentration of negative controls only measured significantly above 0 in overnight reactions, and never measured above 0.5 μM. Reactions and standard curve samples were prepared in triplicate. Reactions were quenched at various time points to obtain at least 3 time points per LpxB variant within the early, linear product accumulation range of the reactions. Specific activities with standard errors and 95% confidence intervals were obtained by linear regression analysis of these data in Graphpad Prism v7.03.

**Differential scanning fluorimetry**. DSF was performed with SYPRO Orange (Life Technologies) and soluble versions of LpxB diluted into soluble LpxB storage buffer to final concentrations of 80 × SYPRO Orange (1.6% dimethyl sulfoxide) and 0.8 mg ml$^{-1}$ LpxB. Aliquots (40 μl) were heated from 20 to 95 °C on a C1000 Touch Thermal Cycler and fluorescence was measured with the FRET channel of a CFX96 Real-Time System (Bio-Rad). The step size of the temperature ramp was 0.5 °C, and the time at each temperature before fluorescence scanning was 30 s. Melting curves were analyzed in Bio-Rad's CFX Manager software to get the melting temperatures ($T_m$) of the LpxB mutants.

**Molecular docking**. UDP-DAG was docked into the LpxB active site cleft with AutoDock Vina 1.1.2[56]. The apo LpxB7S structure (PDB: 5W8S) and the UDP-DAG molecule were prepared for docking in AutoDockTools[67,68] from the MGLTools 1.5.6 suite. For docking, the active site was defined as a box centered at coordinates (− 36.232, 20.151, and 12.865) with x, y, z lengths of 24, 36, and 32 Å. Exhaustiveness was set to 8. Nine models were output.

**Data availability**. The coordinates and structure factors of the crystals structures generated from this research are available at the Protein Data Bank under accession numbers 5W8N (LpxB6S), 5W8S (LpxB7S), and 5W8X (LpxB7S bound to UDP). All other relevant data are available in this article and its Supplementary Information files, or from the corresponding author upon request.

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

## Acknowledgements

We thank Surajit Banerjee for his outstanding assistance during diffraction data collection at the Advanced Photon Source. This work is based upon research conducted at the Northeastern Collaborative Access Team beamlines, which are funded by the National Institute of General Medical Sciences from the National Institutes of Health (P41 GM103403). The Pilatus 6M detector on 24-ID-C beam line is funded by a NIH-ORIP HEI grant (S10 RR029205). This research used resources of the Advanced Photon Source, a U.S. Department of Energy (DOE) Office of Science User Facility operated for the DOE Office of Science by Argonne National Laboratory under Contract No. DE-AC02-06CH11357. We thank Louis E. Metzger IV for his consultation on measuring the activity of LpxB including the purification of UDP-DAG and lipid X, and the proper method of charring TLC plates for visualization of lipids. We thank Jayakanth Kankanala and Zhengquiang Wang for their help with mass spectrometry. UCSF Chimera was utilized to generate 3D structure figures and for protein structure comparisons. Chimera is developed by the Resource for Biocomputing, Visualization, and Informatics at the University of California, San Francisco (supported by NIGMS P41-GM103311). This work was supported by NIH grant GM118047 to H.A.

## Author contributions

H.O.B. designed and carried out all experiments, except those that were performed by outside facilities and were otherwise noted. K.S. ran crystallization robot. K.S. collected the X-ray diffraction data used to generate the final models. J.K.L. and H.O.B. helped with diffraction data collection. H.O.B. built and refined the final models with help from J.K.L. J.K.L. conceived the project and planned initial cloning, purification, and crystallization of wild-type LpxB. H.A. supervised the project. H.O.B. and H.A. prepared

## Additional information

**Competing interests:** J.K.L. is currently employed by Bristol-Myers Squibb (Redwood City, CA). J.K.L. was not employed by BMS during involvement in this research. The remaining authors declare no competing financial interests.

