## [Peer Review File · Nature Communications]

Reviewers' Comments:

Reviewer #1:

Remarks to the Author:

In this work, Bohl and colleagues report the crystal structure of the Lipid A disaccharide synthase (LpxB) from *Escherichia coli*, (i) in its unliganded form and (ii) in complex with UDP. LpxB is an essential enzyme in the biosynthesis of lipopolysaccharide (LPS), the main component of the outer leaflet of the outer membrane in Gram negative bacteria. LpxB catalyzes the reaction between UDP-2,3-diacyl-glucosamine and Lipid X (2,3-diacyl-glucosamine 1-phosphate) to bring forth Lipid A disaccharide and UDP as leaving group. Remarkably, the authors identified a novel dimerization mechanism - domain swapping - for a GT-B enzyme, which is essential to generate a catalytically competent active site. Please find below some comments/suggestions for improvement:

1. Page 2. "This article presents the first crystal structures of a soluble variant of *E. coli* LpxB in the apo form and bound to UDP-N-acetyl-glucosamine (UDP-GlcNAc)". Please replace by "This article presents the first crystal structures of a soluble variant of *E. coli* LpxB in the apo form and bound to UDP." UDP is the only portion of UDP-GlcNAc resolved in the electron density map.

2. Page 3. "While the tertiary structure of LpxB is typical for glycosyltransferase-B (GT-B) family, the quaternary structure is unique." Please add references:

- Albesa-Jové D, Giganti D, Jackson M, Alzari PM, Guerin ME. Structure-function relationships of membrane-associated GT-B glycosyltransferases. *Glycobiology*. 2014 Feb;24(2):108-24. Overall review on GT-B membrane associated GTs.

- Lairson LL, Henrissat B, Davies GJ, Withers SG. Glycosyltransferases: structures, functions, and mechanisms. *Annu Rev Biochem*. 2008;77:521-55. Overall review on GTs.

3. Page 3. "UDP-N-Acetyl-glucosamine Binding". Two options: 1. Please replace "UDP-N-Acetyl-glucosamine Binding" by "UDP-N-Acetyl-glucosamine Binding Site". Please discuss the UDP binding site and then the putative GlcNAc binding site – carefully describing how this information is generated – molecular docking? - and introduce methods accordingly. 2. Please replace "UDP-N-Acetyl-glucosamine Binding" by "UDP Binding Site". Please discuss the putative GlcNAc binding site in the Discussion section – carefully describing how this information is generated – molecular docking - and introduce methods accordingly

4. Page 3. "Only the UDP portion of the molecule is resolved in the electron density map (Figure S2) suggesting that the GlcNAc moiety is flexible." Hydrolysis of UDP-GlcNAc by LpxB is another plausible alternative, as observed in other GTs. Please introduce this concept here and references accordingly.

5. Page 5. "Oligomeriztion" to "Oligomerization"

6. Page 6. "However, there are also important differences between UDP-GlcNAc binding to LpxB and MurG." Please look at point 4.

7. Page 8. "Therefore, the data support the hypothesis that membrane surface-active GT-B enzymes require surface-exposed hydrophobic and basic residues to extract their lipid substrates from the membrane." Please introduce references here:

- Albesa-Jové D, Giganti D, Jackson M, Alzari PM, Guerin ME. Structure-function relationships of membrane-associated GT-B glycosyltransferases. *Glycobiology*. 2014 Feb;24(2):108-24. Overall review on GT-B membrane associated GTs.

- Rodrigo-Unzueta A, Martínez MA, Comino N, Alzari PM, Chenal A, Guerin ME. Molecular Basis of

Membrane Association by the Phosphatidylinositol Mannosyltransferase PimA Enzyme from Mycobacteria. *J. Biol. Chem.* 2016 Jul 1;291(27):13955-63.

8. Page 8. Please consider to replace “suggests that LpxB fully extracts its lipid substrates from the membrane.” by “suggests that LpxB partially or fully extracts its lipid substrates from the membrane.”

9. Page 9. Please consider to replace “the mode of membrane association of surface-active peripheral membrane proteins such as LpxB, PimA, and MurG” by “the mode of membrane association of surface-active membrane proteins such as LpxB, PimA, and MurG”. Some of these GTs could certainly be monotopic *in vivo* according to the following classification:

Blobel, Intracellular protein topogenesis., *Proc. Natl. Acad. Sci. U. S. A.* 77 (1980) 1496–1500.

10. Figures:

- Figure 1: (i) “LpxB is a glycosyltransferase in the Raetz (lipid A synthesis) pathway that catalyzes nucleophilic attack of the 6'-hydroxyl of lipid X on the anomeric carbon of UDP-diacyl-glucosamine (UDP-DAG) to form $\beta(1-6)$ -tetraacyl-disaccharide 1-phosphate (lipid A disaccharide)”. Please include the reaction as Figure 1a. (ii) Please introduce a figure of the isolated monomer - could contribute to clarity. Supplementary figure describing more explicitly the domain swapping and 2D secondary structure cartoon representation could be useful too.

- Figure 2: (i) GlcNAc must be modelled as “chair” configuration and not the unfavorable “boat” configuration. Please see also Figure S2 and point 4 of this report; (ii) Electron density maps – Figure S2 - should be moved to Figure 2.

- Figure 3: (i) Please replace the title “Other important residues”; (ii) Please state the color code used in the legend; (iii) a supplementary figure showing the structural alignment between LpxB, MurG and PimA, the amphipathic alpha-helices, the last C-terminus three alpha helices and the location of single point mutants in LpxB.

11. Tables

- Table 1: Please include experimental phasing data.

- Table 3: please remove H118 as catalytic base. It is incorrect.

I support publication of the manuscript in the case all these questions can be addressed satisfactorily.

Reviewer #2:

Remarks to the Author:

In this manuscript, Bohl and co-workers report the crystal structure of the lipid A disaccharide synthase LpxB from *E. coli*. LpxB is an essential enzyme in the lipid A biosynthetic (Raetz) pathway and is a potential antibiotic target. It is also a challenging target due to its poor biochemical property (e.g., prone to sample aggregation). Based on homology modeling, the authors have identified a cluster of surface hydrophobic residues and have created a 6-residue mutant (6S) and a 7-residue mutant (7S) to improve the sample property. Using these cleverly engineered constructs, the authors were able to solve the LpxB structure, showing that it contains an overall similar tertiary structure to other enzymes of the glycosyltransferase-B family, such as MurG. However, LpxB has distinct structural features such as a highly intertwined C-terminal swapped domain organization.

Despite the high-quality of the crystal structures, the clever protein engineering that has enabled

the structural analysis unfortunately resulted in a complete loss of the LpxB enzymatic activity. In contrast to the authors' statement of maintaining the functional activity, the 6S and 7S LpxB mutants are completely inactive in commonly used detergents (such as Triton X-100); furthermore, these mutants do not functionally substitute LpxB in *E. coli*, suggesting that the engineered enzyme does not retain activity in the native lipid environment. The lack of enzymatic activity raises concerns about the functional relevance of the structure, undermining the impact of the structural analysis.

In addition, the quality of the biochemical assay is poor, and there is no error estimation. In general, the charred TLC assay is not quantitative, and (nearly) all of the bands are very faint and of poor quality. This type of assay may be sufficient to separate an active LpxB enzyme from an inactive mutant, but it is inadequate for quantitative comparisons such as those presented in Table 2 and in Figure 4. The poor quality of the biochemical assay undermines some of the structural interpretations, such as the functional role of the intertwined domain-swapped dimeric structure based on differential enzymatic activity from a mixture of different ratios of LpxB mutants.

Other minor issues include:

- (a) It is suggested that the LpxB reaction is accompanied by a large conformational switch. However, no structural model is presented to explain how it would happen.
- (b) Due to the lack of electron density around the N-acetylglucosamine moiety, it is not clear how this portion of the molecule was modeled.
- (c) The manuscript could also benefit from careful proofreading. For example, in Figure S3, the label "...disaccharide" is misspelled.

Overall, the authors have presented a clever structural solution that has enabled the resolution of the LpxB structure. However, the disconnection of the mutant enzymes from the catalytic activity of LpxB and the poor quality of the enzymatic assays raise concerns about the functional interpretations of the structure observations. As such, this work in its current format may be more suitable for publication in a specialized structural journal.

Reviewer #1 (Remarks to the Author):

In this work, Bohl and colleagues report the crystal structure of the Lipid A disaccharide synthase (LpxB) from Escherichia coli, (i) in its unliganded form and (ii) in complex with UDP. LpxB is an essential enzyme in the biosynthesis of lipopolysaccharide (LPS), the main component of the outer leaflet of the outer membrane in Gram negative bacteria. LpxB catalyzes the reaction between UDP-2,3-diacyl-glucosamine and Lipid X (2,3-diacyl-glucosamine 1-phosphate) to bring forth Lipid A disaccharide and UDP as leaving group. Remarkably, the authors identified a novel dimerization mechanism - domain swapping - for a GT-B enzyme, which is essential to generate a catalytically competent active site. Please find below some comments/suggestions for improvement:

1. Page 2. *“This article presents the first crystal structures of a soluble variant of E. coli LpxB in the apo form and bound to UDP-N-acetyl-glucosamine (UDP-GlcNAc)”*. Please replace by *“This article presents the first crystal structures of a soluble variant of E. coli LpxB in the apo form and bound to UDP.”* UDP is the only portion of UDP-GlcNAc resolved in the electron density map.

This change was made.

2. Page 3. *“While the tertiary structure of LpxB is typical for glycosyltransferase-B (GT-B) family, the quaternary structure is unique.”* Please add references:

- Albesa-Jové D, Giganti D, Jackson M, Alzari PM, Guerin ME. Structure-function relationships of membrane-associated GT-B glycosyltransferases. *Glycobiology*. 2014 Feb;24(2):108-24. Overall review on GT-B membrane associated GTs.

- Lairson LL, Henrissat B, Davies GJ, Withers SG. Glycosyltransferases: structures, functions, and mechanisms. *Annu Rev Biochem*. 2008;77:521-55. Overall review on GTs.

These references were added.

3. Page 3. *“UDP-N-Acetyl-glucosamine Binding”*. Two options: 1. Please replace *“UDP-N-Acetyl-glucosamine Binding”* by *“UDP-N-Acetyl-glucosamine Binding Site”*. Please discuss the UDP binding site and then the putative GlcNAc binding site – carefully describing how this information is generated – molecular docking? - and introduce methods accordingly. 2. Please replace *“UDP-N-Acetyl-glucosamine Binding”* by *“UDP Binding Site”*. Please discuss the putative GlcNAc binding site in the Discussion section – carefully describing how this information is generated – molecular docking - and introduce methods accordingly.

We changed the section title to *“UDP Binding Site”*. UDP-GlcNAc was only placed with the PHENIX x-ray data refinement software. While this does include geometry restraints, the software attempts to fit the molecule to the electron density. The density for the UDP portion is strong, so we have high confidence in the position of UDP. However, there was no density

visible for GlcNAc when the electron density map was at 1.00 rmsd. Therefore, we have little confidence in the position or even presence of GlcNAc. We used UDP-GlcNAc during refinement despite this because we found that deleting GlcNAc increased the R factors and produced a small amount of positive difference density. However, as Reviewer 1 brought to our attention that nucleotide-charged sugars are often hydrolyzed during soaking with glycosyltransferase crystals, we reexamined the structure with GlcNAc removed. We decided that the difference density could easily be caused by the presence of an ion or an alternate conformation of the β -phosphate with low occupancy. Therefore, we re-refined the structure with only UDP bound. We replaced the coordinates in the PDB with the UDP-bound structure and updated the figures in the paper to show UDP. We still included discussion of the differences in the positions of structural elements involved in binding GlcNAc in MurG and in the positions of these elements in LpxB.

4. *Page 3. "Only the UDP portion of the molecule is resolved in the electron density map (Figure S2) suggesting that the GlcNAc moiety is flexible." Hydrolysis of UDP-GlcNAc by LpxB is another plausible alternative, as observed in other GTs. Please introduce this concept here and references accordingly.*

We mentioned this possibility and referenced other studies where the nucleotide-charged sugar was cleaved during soaking or co-crystallization with glycosyltransferases.

5. *Page 5. "Oligomeriztion" to "Oligomerization"*

This typo was fixed.

6. *Page 6. "However, there are also important differences between UDP-GlcNAc binding to LpxB and MurG." Please look at point 4.*

We altered this section of the Discussion to describe differences in the binding of UDP, differences in the placement of the helix that is involved in contacting UDP-GlcNAc in MurG, and the implications for UDP-DAG binding in LpxB.

7. *Page 8. "Therefore, the data support the hypothesis that membrane surface-active GT-B enzymes require surface-exposed hydrophobic and basic residues to extract their lipid substrates from the membrane." Please introduce references here:*

- *Albesa-Jové D, Giganti D, Jackson M, Alzari PM, Guerin ME. Structure-function relationships of membrane-associated GT-B glycosyltransferases. Glycobiology. 2014 Feb;24(2):108-24. Overall review on GT-B membrane associated GTs.*

- *Rodrigo-Unzueta A, Martínez MA, Comino N, Alzari PM, Chenal A, Guerin ME. Molecular Basis*

of Membrane Association by the Phosphatidylinositol Mannosyltransferase PimA Enzyme from Mycobacteria. J. Biol. Chem. 2016 Jul 1;291(27):13955-63.

These references were added.

8. *Page 8. Please consider to replace “suggests that LpxB fully extracts its lipid substrates from the membrane.” by “suggests that LpxB partially or fully extracts its lipid substrates from the membrane.”*

This was changed to “mostly or fully”. In addition, we added to the Discussion analysis of molecular docking models for the natural donor substrate of LpxB (UDP-diacyl-glucosamine).

9. *9. Page 9. Please consider to replace “the mode of membrane association of surface-active peripheral membrane proteins such as LpxB, PimA, and MurG” by “the mode of membrane association of surface-active membrane proteins such as LpxB, PimA, and MurG”. Some of these GTs could certainly be monotopic in vivo according to the following classification:*

Blobel, Intracellular protein topogenesis., Proc. Natl. Acad. Sci. U. S. A. 77 (1980) 1496–1500.

This change was made.

10. *10. Figures:*

- Figure 1: (i) “LpxB is a glycosyltransferase in the Raetz (lipid A synthesis) pathway that catalyzes nucleophilic attack of the 6'-hydroxyl of lipid X on the anomeric carbon of UDP-diacyl-glucosamine (UDP-DAG) to form $\beta(1-6)$ -tetraacyl-disaccharide 1-phosphate (lipid A disaccharide)”. Please include the reaction as Figure 1a. (ii) Please introduce a figure of the isolated monomer - could contribute to clarity. Supplementary figure describing more explicitly the domain swapping and 2D secondary structure cartoon representation could be useful too.

- (i) The reaction is now included as Figure 1.
- (ii) The isolated monomer and the 2D representation of the dimerization were combined as Figure S2.

- Figure 2: (i) GlcNAc must be modelled as “chair” configuration and not the unfavorable “boat” configuration. Please see also Figure S2 and point 4 of this report; (ii) Electron density maps – Figure S2 - should be moved to Figure 2.

- (i) The GlcNAc was removed as described above.
- (ii) The electron density map for UDP was added to this figure.

- Figure 3: (i) Please replace the title “Other important residues”; (ii) Please state the color code used in the legend; (iii) a supplementary figure showing the structural alignment between LpxB, MurG and PimA, the amphipathic alpha-helices, the last C-terminus three alpha helices and the location of single point mutants in LpxB.

(i) The title was changed to “Interactions that stabilize the dimeric structure.” The comparison of the locations of H19 and D98 was moved to Figure 6, and additional interactions that stabilize the position of the N-terminal domain were added to the figure in question.

(ii) We assume this refers to Figure 4. We included the colors used for the proteins in the legend.

(iii) The structural alignment was added as Figure S7.

11. Tables

- Table 1: Please include experimental phasing data.

The diffraction statistics for the selenomethionine-derivative crystal used for experimental phasing were already included under “LpxB7S Sem” in Table 1. We attempted to make this more explicit by naming the column “LpxB7S SeMet (SAD data)”.

- Table 3: please remove H118 as catalytic base. It is incorrect.

This was removed from Table 2.

I support publication of the manuscript in the case all these questions can be addressed satisfactorily.

Reviewer #2 (Remarks to the Author):

In this manuscript, Bohl and co-workers report the crystal structure of the lipid A disaccharide synthase LpxB from E. coli. LpxB is an essential enzyme in the lipid A biosynthetic (Raetz) pathway and is a potential antibiotic target. It is also a challenging target due to its poor biochemical property (e.g., prone to sample aggregation). Based on homology modeling, the authors have identified a cluster of surface hydrophobic residues and have created a 6-residue mutant (6S) and a 7-residue mutant (7S) to improve the sample property. Using these cleverly engineered constructs, the authors were able to solve the LpxB structure, showing that it contains an overall similar tertiary structure to other enzymes of the glycosyltransferase-B family, such as MurG. However, LpxB has distinct structural features such as a highly intertwined C-terminal swapped domain organization.

Despite the high-quality of the crystal structures, the clever protein engineering that has enabled the structural analysis unfortunately resulted in a complete loss of the LpxB enzymatic activity. In contrast to the authors’ statement of maintaining the functional activity, the 6S and 7S LpxB mutants

are completely inactive in commonly used detergents (such as Triton X-100); furthermore, these mutants do not functionally substitute LpxB in E. coli, suggesting that the engineered enzyme does not retain activity in the native lipid environment. The lack of enzymatic activity raises concerns about the functional relevance of the structure, undermining the impact of the structural analysis.

There is sufficient evidence that the mutations made have not damaged the active site or overall structure. First, while “catalytically competent” may be better terminology than “catalytically active”, we showed in Figure S3 that LpxB6S retains the ability to form lipid A disaccharide under the right conditions. Second, we were able to compare the relative activities double and triple mutants of these hydrophobic residues, and we found that activity correlated with the hydrophobicity of the mutated region, which showed that no single residue among the mutated 6 was critical. Third, the role of this region in membrane-association, rather than formation of the active site, is supported by homology with the well-studied enzyme, PimA. Overlaying the N-terminal domains of LpxB and PimA (shown in Figure 5 (now 6) and now also in structural alignment form in Figure S7) identified the surface exposed hydrophobic loop-amphipathic helix as a conserved motif. We referenced Guerin *et al.* 2007, which showed that deletion of the hydrophobic loop or mutation of the basic residues in the amphipathic helix eliminated activity of PimA¹. We now add a reference providing more evidence that the helix is involved in membrane association in PimA². While this motif is not as similar in MurG, examination of the structural alignment shows that MurG has a similar amphipathic helix at this position. Therefore, we contend that the mutations made in LpxB6S have merely disrupted its ability to associate with the membrane, and we hypothesize with confidence that the active site and overall structure of LpxB6S are the same as for the wild-type enzyme. Thus, this structure remains highly relevant to areas of research including lipid A synthesis, glycosyltransferase enzymology, and antibiotic development.

1. Guerin, M. E. *et al.* Molecular recognition and interfacial catalysis by the essential phosphatidylinositol mannosyltransferase PimA from mycobacteria. *J. Biol. Chem.* **282**, 20705–20714 (2007).
2. Rodrigo-Unzueta, A. *et al.* Molecular Basis of Membrane Association by the Phosphatidylinositol Mannosyltransferase PimA Enzyme from Mycobacteria. *J. Biol. Chem.* **291**, 13955–13963 (2016).

In addition, the quality of the biochemical assay is poor, and there is no error estimation. In general, the charred TLC assay is not quantitative, and (nearly) all of the bands are very faint and of poor quality. This type of assay may be sufficient to separate an active LpxB enzyme from an inactive mutant, but it is inadequate for quantitative comparisons such as those presented in Table 2 and in Figure 4. The poor quality of the biochemical assay undermines some of the structural interpretations, such as the functional role of the intertwined domain-swapped dimeric structure based on differential enzymatic activity from a mixture of different ratios of LpxB mutants.

We now provide further evidence that corroborates these data by an alternative assay and that validates our interpretation of TLC data. We now include data (Figure 4) quantifying the amount of UDP released at different time points using a UDP-Glo Glycosyltransferase Assay kit (Promega), which allows UDP concentration to be quantified with luciferase activity using a standard curve. These data support the relative activities presented in Table 2 and Figure 4. The only changes were that the enhanced

sensitivity of this assay allowed us to see the decrease in activity of the L72SL76S and L75SL76S double mutants, which were indistinguishable from wild-type LpxB by the TLC-based assay, and the slight increase in activity of the 1:9 mixture of LpxBFN and LpxBR201A, which was indistinguishable from LpxBFN by TLC. Since we were able to do proper replicates with this kit, these data are presented with error bars (99% confidence intervals, $n=3$ except in two cases where $n=2$). In addition, we now include evidence that our TLC charring method is quantitative. Somewhat different methods of TLC charring have been previously reported as methods for lipid quantification^{3,4}. To validate our method, we now include standard TLC plates as Figure S10, one with different amounts of the reactants run and one with different amounts of the reactants reacted to completion with lipid X in excess. We analyzed the intensity of the bands in ImageJ software and plotted the results showing a linear relationship between intensity and lipid amount for both reactants and the lipid A disaccharide product. Thus, the standard plates show that changes in intensity, particularly large changes, can be confidently interpreted as changes in lipid amount, and this validates our interpretation of Figures 4 and S3. ImageJ analysis for these plates has now been added as Figures S3 and S5. These data still lack error bars because the size of the TLC plates and the amount of reagents required prevented us from running standard curves (for absolute quantification) on each TLC plate. Thus, we cannot directly compare the amounts of lipids in replicate reactions run on separate TLC plates. However, LpxB6S activity in 0.9 M NDSB 201 was replicated once, and the increase in activity of the 1:2 mixture of LpxBFN and LpxBR201A was replicated 4 times. TLC assays for lower ratios of LpxBFN were not repeated. We have opted to remove the stabilization at 4°C/active protein pull-down plates from Figures 4 and S3 as these plates were more difficult to interpret, and we judged them as unimportant for demonstrating the formation of the swapped dimer in light of our UDP-release data, which strongly supports the formation of a complementary dimer between LpxBFN and LpxBR201A.

In the case of Table 2, these semi-quantitative data did not require comparison of band intensities. These data were based only on the ability to detect the lipids by charring. Reactions were stated to have “reached completion” when the product was visible and the limiting reagent (UDP-DAG) was no longer visible. Reactions were run at various time points, and the end points of the ranges presented in Table 2 are the maximum time when the reaction was observed to have not reached completion and the minimum time when the reaction was observed to have reached completion. The one case where these data were supplemented by observation of relative intensities is in the note about the F298EN316A mutant, which required comparison to band intensities from complete (wild-type) and negative control (R201A) reactions run on the same plate. Therefore, Table 2 provides a semi-quantitative comparison of enzyme activity, which is now corroborated by UDP-release quantification (Figure 4). We did not intend for the data in Table 2 to be viewed as quantitative data, such as specific activities, and we have added more detail about the generation of these data to the Online Methods.

3. Rodríguez, S., Cesio, M. V., Heinzen, H. & Moyna, P. Determination of the phospholipid/lipophilic compounds ratio in liposomes by thin-layer chromatography scanning densitometry. *Lipids* **35**, 1033–1036 (2000).
4. Schariter, J. A., Pachuski, J., Fried, B. & Sherma, J. Determination of Neutral Lipids and Phospholipids in the Cercariae of *Schistosoma Mansoni* by High Performance Thin Layer Chromatography. *J. Liq. Chromatogr. Relat. Technol.* **25**, 1615–1622 (2002).

Other minor issues include:

(a) It is suggested that the LpxB reaction is accompanied by a large conformational switch. However, no structural model is presented to explain how it would happen.

We had suggested that LpxB undergoes a motion similar to MurG, which showed a $\sim 10^\circ$ hinge-like movement from the apo to the UDP-GlcNAc-bound structure. We now cite other GT-B enzymes where this closing motion was observed and discuss the possibility of this movement in LpxB.

(b) Due to the lack of electron density around the N-acetylglucosamine moiety, it is not clear how this portion of the molecule was modeled.

Please see response under comment 3 by Reviewer 1.

(c) The manuscript could also benefit from careful proofreading. For example, in Figure S3, the label "...disaccharide" is misspelled.

This typo was fixed.

Overall, the authors have presented a clever structural solution that has enabled the resolution of the LpxB structure. However, the disconnection of the mutant enzymes from the catalytic activity of LpxB and the poor quality of the enzymatic assays raise concerns about the functional interpretations of the structure observations. As such, this work in its current format may be more suitable for publication in a specialized structural journal.

Reviewers' Comments:

Reviewer #1:

Remarks to the Author:

The authors have taken care of my comments and suggestions very satisfactorily, and I support the publication of the manuscript in Nature Communications.

Reviewer #2:

Remarks to the Author:

In this revised manuscript, Borl and co-workers report the crystal structure of a solubilized, mutated form of the lipid A disaccharide synthase LpxB from *E. coli*. The revised manuscript shows significant improvement over the original manuscript. It is regretful, however, that the authors half-heartedly addressed the request for more rigorous enzymatic assays, which are critical to support their structural model.

The charred TLC assay is not a suitable quantitative assay because it requires internal standards for calibration and because it is impossible to achieve uniform charring. As a result, the background and the intensities of the sample bands are highly variable. Such an effect is visually evident from the different charring levels of the background areas at different locations of the TLC plate. The gold standard of the LpxB activity assay is the ³²P-based autoradiographic TLC assay reported by Metzger and Raetz in the initial characterization of LpxB (Biochemistry 2009, 48, 11559–11571). It is not clear why the authors elect not to use this well-established assay. That being said, the coupled UDP assay is an excellent substitute of the ³²P assay, and this reviewer is willing to accept it as a suitable substitute of the ³²P autoradiographic TLC assay. However, despite having developed an excellent enzymatic assay, the authors elected to retain much of the non-quantitative assay results to describe the effects of the LpxB mutants, which have created a lot of questions and inconsistencies.

For example, the activity of the *E. coli* LpxB N316A mutant was reported to be 0.1% of the WT enzyme activity by Metzger and Raetz (Biochemistry 2009, 48, 11559–11571), while its activity in the UDP coupled assay appeared to be much higher (Figure 4A). A close examination shows that the Metzger enzymatic activity was reported as the enzyme specific activity, which is typically determined from the slope of the linear product accumulation curve to calculate the rate of the substrate turn over. In order to maintain the linear product accumulation, typically only the data points below 20% substrate conversion are used for calculating the enzyme specific activity. However, it is evident from the time course in Figure 4A that under the assay conditions used here, the substrate has been completely depleted for the WT LpxB enzyme, therefore precluding a proper calculation of the enzyme specific activity and a proper comparison of the specific activity of the LpxB N316A mutant from this study with that from the previous study.

It is also not clear why the authors didn't include the various time points for some of the mutants in Figure 4A. For example, the mutants 7276 and 7576 were reported to reach reaction completion in less than 5 minutes in Table 2. Yet, the results from Figure 4A indicate that these reactions only reached ~50% substrate conversion in comparison to the wild-type enzyme in 10 minutes!

Likewise, the "Time to Reaction Completion" is totally inappropriate for measuring and comparing enzymatic activities. These numbers should be replaced with enzyme specific activities measured using the rates of linear production accumulation calculated using data from less than 20% substrate turn over.

Along the same line, the authors need to quantify the residual enzymatic activity of the 6S LpxB mutant in detergent and in NDSB 201, compare them with the WT LpxB enzyme, and specifically list these results in Table 2.

It is critical for the authors to obtain high quality enzymology data to support their structural observations for several reasons. First, the same Metzger paper (Biochemistry 2009, 48, 11559–11571) referenced by the authors also reported the observation of a monomeric state of *H. influenzae* LpxB based on the size-exclusion chromatography data, which was not discussed here; second, the authors' analytical centrifugation data did not yield a convergent MW; third, the authors' more sensitive UDP coupled assay shows that the optimal enzymatic activity for the mixture of the LpxB-FN (F298E/N316A) double mutant with the LpxB-R201A mutant is obtained at the 1:2 molar ratio. However, one would have certainly predicted that the maximum activity should be obtained at the 1:1 ratio based on the statistical analysis of their dimeric model.

Perhaps many of these abnormalities would go away when the authors report the appropriate enzyme specific activity.

Following on the LpxB F298E/N316A double mutant, it is not clear why the authors presented the result of the N316A mutant, but not the F298E mutant. As the N316A mutant is not directly involved in the UDP binding, it is not clear why the authors chose not to use the single point mutant of F298 (e.g., F298E), which directly interacts with R201 for UDP interaction. The results from these tests (the LpxB F298E mutant alone and the LpxB F298E mutant mixture with the LpxB R201A mutant) would have provided stronger support for the authors' structural model.

The authors also need to explain their choice of the F298E mutation. In their structural model, F298 forms a cation-pi stacking with the side chain of R201. It is not clear if the F298E mutation will effectively disrupt the F298-R201 interaction. It seemed that a more suitable mutation would be F298R or F298K for charge repulsion or F298A for elimination of the sidechain interaction.

Finally, in the revised Table 1, the outer shells statistics for R_{work} (0.3512) and R_{free} (0.3278) for the LpxB7S + UDP complex might be reversed. The authors should double check their numbers.

To summarize, despite the significant improvement over the original manuscript, this revised manuscript falls short in its enzymatic assays. Given that the authors have already purified the relevant enzymes and have developed the appropriate assays, it would be a straightforward excise for the authors to obtain quantitative enzymatic data. Without the proper measurements of the enzymatic specific activities, this manuscript is unfit for publication in Nature Communications.

Reviewer #1 (Remarks to the Author):

The authors have taken care of my comments and suggestions very satisfactorily, and I support the publication of the manuscript in Nature Communications.

Reviewer #2 (Remarks to the Author):

*In this revised manuscript, Borl and co-workers report the crystal structure of a solubilized, mutated form of the lipid A disaccharide synthase LpxB from *E. coli*. The revised manuscript shows significant improvement over the original manuscript. It is regretful, however, that the authors half-heartedly addressed the request for more rigorous enzymatic assays, which are critical to support their structural model.*

The charred TLC assay is not a suitable quantitative assay because it requires internal standards for calibration and because it is impossible to achieve uniform charring. As a result, the background and the intensities of the sample bands are highly variable. Such an effect is visually evident from the different charring levels of the background areas at different locations of the TLC plate. The gold standard of the LpxB activity assay is the ³²P-based autoradiographic TLC assay reported by Metzger and Raetz in the initial characterization of LpxB (Biochemistry 2009, 48, 11559–11571). It is not clear why the authors elect not to use this well-established assay. That being said, the coupled UDP assay is an excellent substitute of the ³²P assay, and this reviewer is willing to accept it as a suitable substitute of the ³²P autoradiographic TLC assay. However, despite having developed an excellent enzymatic assay, the authors elected to retain much of the non-quantitative assay results to describe the effects of the LpxB mutants, which have created a lot of questions and inconsistencies.

*For example, the activity of the *E. coli* LpxB N316A mutant was reported to be 0.1% of the WT enzyme activity by Metzger and Raetz (Biochemistry 2009, 48, 11559–11571), while its activity in the UDP coupled assay appeared to be much higher (Figure 4A). A close examination shows that the Metzger enzymatic activity was reported as the enzyme specific activity, which is typically determined from the slope of the linear product accumulation curve to calculate the rate of the substrate turn over. In order to maintain the linear product accumulation, typically only the data points below 20% substrate conversion are used for calculating the enzyme specific activity. However, it is evident from the time course in Figure 4A that under the assay conditions used here, the substrate has been completely depleted for the WT LpxB enzyme, therefore precluding a proper calculation of the enzyme specific activity and a proper comparison of the specific activity of the LpxB N316A mutant from this study with that from the previous study.*

It is also not clear why the authors didn't include the various time points for some of the mutants in Figure 4A. For example, the mutants 7276 and 7576 were reported to reach reaction completion in less than 5 minutes in Table 2. Yet, the results from Figure 4A indicate that these reactions only reached ~50% substrate conversion in comparison to the wild-type enzyme in 10 minutes!

Likewise, the “Time to Reaction Completion” is totally inappropriate for measuring and comparing enzymatic activities. These numbers should be replaced with enzyme specific activities measured using the rates of linear production accumulation calculated using data from less than 20% substrate turn over.

The UDP-Glo assay was used to determine the specific activities of LpxB variants as suggested above. LpxB concentrations and reaction times were adjusted as appropriate to obtain at least 3 time points within the early, linear portion of the reactions, and the specific activities with standard errors and 95% confidence intervals were calculated from these data in Graphpad Prism. These data are presented in Table 2, and all TLC-based data have been moved to the Supplementary Information. Comparison of activities presented in the Results is based on the specific activities in Table 2, and TLC data are only referenced as supporting data. The specific activities support our previous conclusions regarding the importance of the hydrophobic patch and the C-terminal swap.

Along the same line, the authors need to quantify the residual enzymatic activity of the 6S LpxB mutant in detergent and in NDSB 201, compare them with the WT LpxB enzyme, and specifically list these results in Table 2.

The activity of LpxB6S was below the detection limit of this assay. In Triton X-100, the amount of UDP released with 10 μ M LpxB6S was not significantly different from that detected in negative control reactions lacking LpxB even when the reactions were run overnight (>17 hr). (Negative control reactions only had UDP concentrations significantly above 0 in overnight reactions, and these still remained below 0.5 μ M.) In NDSB-201, 10 μ M LpxB6S did produce more UDP than the negative control (0.7 μ M at 5 hr), but there was no increase in UDP concentration from 5 hr to 17 hr. Thus, a specific activity could not be calculated. Therefore, the specific activity of LpxB6S is listed as “Below detection limit” in Table 2. We have provided an updated image of a TLC plate including overnight reaction products of LpxB6S (Figure S3), which provides a clearer view of the lipid A disaccharide band in this lane to show that LpxB6S retains some very weak activity. While we have obtained a greater dynamic range than presented by Metzger and Raetz¹ and were able to measure activities well below 0.01% of wild-type (Table 2), the activity of LpxB6S remained below this range. As the specific activity of the triple mutant LpxBLLL had a very low specific activity (the lowest we could measure), it is not surprising that adding 3 additional mutations to the hydrophobic patch would decrease the activity to below the detection limit.

1. Metzger, L. E., 4th & Raetz, C. R. H. Purification and characterization of the lipid A disaccharide synthase (LpxB) from *Escherichia coli*, a peripheral membrane protein. *Biochemistry (Mosc.)* **48**, 11559–11571 (2009).

*It is critical for the authors to obtain high quality enzymology data to support their structural observations for several reasons. First, the same Metzger paper (Biochemistry 2009, 48, 11559–11571) referenced by the authors also reported the observation of a monomeric state of *H. influenzae* LpxB based on the size-exclusion chromatography data, which was not discussed here; second, the authors’ analytical centrifugation data did not yield a convergent MW; third, the authors’ more sensitive UDP coupled assay shows that that the optimal enzymatic activity for the mixture of the LpxB-FN (F298E/N316A) double mutant with the LpxB-R201A mutant is obtained at the 1:2 molar ratio. However,*

one would have certainly predicted that the maximum activity should be obtained at the 1:1 ratio based on the statistical analysis of their dimeric model.

While Metzger and Raetz described *H influenzae* LpxB as an apparent monomer, they describe the *E. coli* LpxB as an apparent octamer that was converted to an apparent dimer when DDM detergent was added. This inconsistency in addition to our own size exclusion data prompted us to utilize analytical ultracentrifugation to distinguish between the dimeric and monomeric states. These data, while not ideal, were more consistent with a dimer. We now mention the results of Metzger and Raetz for *E. coli* LpxB as part of the justification for the use of AUC. As for *H. influenzae* LpxB, we agree that it is very unlikely that the oligomerization of this LpxB is different. However, the HiLpxB dimer may not have been stable under the size exclusion conditions used as Metzger and Raetz note that “The concentration of NaCl in the *H. influenzae* LpxB size exclusion chromatography buffer was maintained at 500 mM, because the purified enzyme, when concentrated to >5 mg/mL, precipitated at lower ionic strength,” whereas they performed size exclusion of 5 mg/mL EcLpxB in 200 mM NaCl. Furthermore, size exclusion chromatography depends on the hydrodynamic radius of the protein thus giving skewed molecular mass estimations for proteins with irregular shapes. Thus, we concluded that size exclusion chromatography was an unreliable method for determining the molecular weight of LpxB and utilized AUC to measure the equilibrium sedimentation of LpxB, which is a more reliable measure of molecular weight².

2. Cole, J. L., Lary, J. W., P. Moody, T. & Laue, T. M. Analytical Ultracentrifugation: Sedimentation Velocity and Sedimentation Equilibrium. in *Methods in Cell Biology* **84**, 143–179 (Academic Press, 2008).

Perhaps many of these abnormalities would go away when the authors report the appropriate enzyme specific activity.

This discrepancy did disappear when specific activities were measured. The 50% LpxBFN mixture shows the highest specific activity of the LpxBR201A-LpxBFN mixtures (Table 2).

Following on the LpxB F298E/N316A double mutant, it is not clear why the authors presented the result of the N316A mutant, but not the F298E mutant. As the N316A mutant is not directly involved in the UDP binding, it is not clear why the authors chose not to use the single point mutant of F298 (e.g., F298E), which directly interacts with R201 for UDP interaction. The results from these tests (the LpxB F298E mutant alone and the LpxB F298E mutant mixture with the LpxB R201A mutant) would have provided stronger support for the authors' structural model.

An explanation for the choice of the F298E mutation and the use of the F298E/N316A double mutant has been added to the ‘Oligomerization of LpxB in Solution’ section of the Results. Our intention in adding the F298E mutation to the LpxB N316A mutant was to obtain a derivative with mutations only in the C-terminal swapped portion of the protein that would be sufficiently inactive to observe a rapid and significant increase in activity when mixed with LpxBR201A. We measured the specific activity of LpxBN316A at 0.3% of wild-type. However, we failed to observe any increase in specific activity when a 50% LpxBN316A mixture was made with LpxBR201A. This is most likely because the change in activity upon formation of this mixture depends on the kinetics of dimer exchange, which are likely slow for such an extensive dimerization interface. We could have elected to leave the protein mixtures overnight to allow equilibration of dimer exchange, but we were concerned about observed losses in LpxB activity when stored at 4°C that were particularly noticeable for poorly active variants. Thus, we opted to test all activities with freshly thawed enzyme, so a very inactive enzyme was required for these experiments.

Therefore, we decided to build on the loss of activity of the N316A mutant instead of making another single mutant in hopes that this mutation alone would be sufficient. Regardless of the number of mutations in the C-terminal swapped tail, the ability of a poorly active LpxB with its mutation in the unswapped portion to produce more activity when combined with a poorly active LpxB with its mutations only in the swapped portion, as compared to the activity of either protein alone, supports the formation of heterodimers with one subunit containing none of these mutations. We think these data are very convincing for ruling out the unlikely possibility of a non-physiological C-terminally swapped dimer being formed during crystallization.

The authors also need to explain their choice of the F298E mutation. In their structural model, F298 forms a cation- π stacking with the side chain of R201. It is not clear if the F298E mutation will effectively disrupt the F298-R201 interaction. It seemed that a more suitable mutation would be F298R or F298K for charge repulsion or F298A for elimination of the sidechain interaction.

Our intention was to alter the electrostatics of the active site by tying up the positive charge of R201 with a negative charge rather than to eliminate any interaction between residues 201 and 298. Our logic was that if R201 helps to stabilize the negative charge of the UDP leaving group, pairing this side chain with a negatively charged side chain would inhibit this function by balancing the charge and by altering the conformation of the R201. This explanation for the Phe to Glu mutation was added to the Results as stated above.

Finally, in the revised Table 1, the outer shells statistics for Rwork (0.3512) and Rfree (0.3278) for the LpxB7S + UDP complex might be reversed. The authors should double check their numbers.

These numbers are correct. The magnitudes of the R-factors in the highest shell were the opposite of what is expected. The magnitudes were as expected for all other shells. Excluding the data in the highest resolution shell did not improve any refinement parameters. We suspect that some particularly noisy data were randomly assigned to the working set and excluded from the test set.

To summarize, despite the significant improvement over the original manuscript, this revised manuscript falls short in its enzymatic assays. Given that the authors have already purified the relevant enzymes and have developed the appropriate assays, it would be a straightforward excise for the authors to obtain quantitative enzymatic data. Without the proper measurements of the enzymatic specific activities, this manuscript is unfit for publication in Nature Communications.

Reviewers' Comments:

Reviewer #2:

Remarks to the Author:

In this further revised manuscript, Bohl and co-workers have dramatically improved their enzymatic assays and have adequately addressed reviewer critiques. This further revised manuscript is now suitable for publication in Nature Communications.